# GLIPv2: Unifying Localization and VL Understanding

**Haotian Zhang**[*1†], **Pengchuan Zhang**[*2†♠], **Xiaowei Hu**[3], **Yen-Chun Chen**[3], **Liunian Harold Li**[4†]
**Xiyang Dai**[3], **Lijuan Wang**[3], **Lu Yuan**[3], **Jenq-Neng Hwang**[1], **Jianfeng Gao**[3]
[1]University of Washington, [2]Meta AI, [3]Microsoft, [4]UCLA
{haotiz,hwang}@uw.edu,pengchuanzhang@fb.com,liunian.harold.li@cs.ucla.edu,
{Xiaowei.Hu,Yen-Chun.Chen,Xiyang.Dai,lijuanw,luyuan,jfgao}@microsoft.com

## Abstract

We present GLIPv2, a grounded VL understanding model, that serves both local-
ization tasks (e.g., object detection, instance segmentation) and Vision-Language
(VL) understanding tasks (e.g., VQA, image captioning). GLIPv2 elegantly uni-
fies localization pre-training and Vision-Language Pre-training (VLP) with three
pre-training tasks: phrase grounding as a VL reformulation of the detection task,
region-word contrastive learning as a novel region-word level contrastive learning
task, and the masked language modeling. This unification not only simplifies the
previous multi-stage VLP procedure but also achieves mutual benefits between
localization and understanding tasks. Experimental results show that a single
GLIPv2 model (all model weights are shared) achieves near SoTA performance
on various localization and understanding tasks. The model also shows (1) strong
zero-shot and few-shot adaption performance on open-vocabulary object detection
tasks and (2) superior grounding capability on VL understanding tasks. Code is
released at `https://github.com/microsoft/GLIP`.

## 1 Introduction

Recently, a general interest arises in building general-purpose vision systems [21, 24, 56, 42], also
called vision foundation models [6, 57], that solve various vision tasks simultaneously, such as
image classification [30], object detection [39], and Visual-Language (VL) understanding [3, 11, 27].
Of particular interest, is the unification between *localization* tasks (e.g., object detection [39] and
segmentation [8, 20]) and VL *understanding* tasks (e.g., VQA [3] and image captioning [11]).
Localization pre-training benefits VL tasks [1, 59], and the "localization->VLP" two-stage pre-
training procedure [41, 49, 13, 48, 34, 32, 61, 37, 35] is the common practice in VL community. A
long-standing challenge is the unification of localization and understanding, which aims at *mutual*
benefit between these two kinds of tasks, simplified pre-training procedure, and reduced pre-training
cost.

However, these two kinds of tasks appear to be dramatically different: localization tasks are vision-
only and require fine-grained output (e.g., bounding boxes or pixel masks), while VL understanding
tasks emphasize fusion between two modalities and require high-level semantic outputs (e.g., answers
or captions).

[21, 24, 56] have made early attempts at unifying these tasks in a straightforward multi-task manner,
where a low-level visual encoder is shared across tasks, and two separate high-level branches
are designed for localization and VL understanding, respectively. The localization tasks are still
vision-only and do not benefit from the rich semantics in vision-language data. As a result, such
unified models see the marginal mutual benefit or even performance degradation [24] compared with
task-specific models.

In this paper, we identify "VL grounding" as a "meta"-capability for localization and understanding
capabilities. VL grounding involves not only *understanding* an input sentence but also *localizing*

---

*The two authors contributed equally. †Work done at Microsoft Research. ♠ Corresponding author.

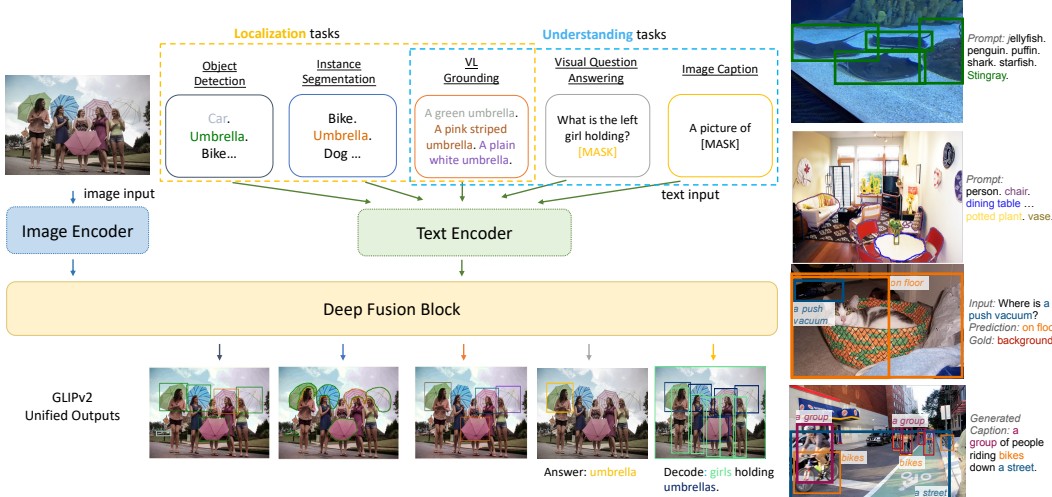

Figure 1: Left: GLIPv2, a pre-trained grounded VL understanding model, unifies various localization and VL understanding tasks. These two kinds of tasks mutually benefit each other, and enables new capabilities such as language-guided detection/segmentation and grounded VQA/captioning. Right: Additional examples from ODinW (detection), LVIS (segmentation), VQA, COCO Captioning.

the mentioned entities in the image (see an example in Figure 1). We build a **grounded VL understanding** model (GLIPv2) as a unified model for localization and VL understanding tasks.

**Localization + VL understanding = grounded VL understanding**. Localization tasks involve both localization and semantic classification, where classification can be cast as a VL understanding problem using the *classification-to-matching* trick (Section 3.1). Therefore, we reformulate localization tasks as VL grounding tasks, in which the language input is a synthesized sentence as the concatenation of category names [36]. Localization data are turned into VL grounding data, accordingly. The massive VL understanding data (image-text pairs) can be easily turned into VL grounding data in a self-training manner [36]. Therefore, GLIPv2 has a unified pre-training process: all task data are turned into grounding data and GLIPv2 is pre-trained to perform grounded VL understanding.

**A stronger VL grounding task: inter-image region-word contrastive learning**. GLIP [36] proposes the phrase grounding task as its pre-training task, which we argue is an easy task and does not fully utilize data information. For example, in the VL grounding task in Figure 1, the phrase grounding task only requires the model to match a given image region to one of the three phrases in the text input, i.e., "green, pink striped, or plain white umbrella?". This 1-in-3 choice is very easy, only requires color understanding, but loses lots of information in this grounding data: the umbrellas are not any other colors, like black, yellow, etc; objects in those regions are umbrellas but not any other categories, like car, bike, etc. From a contrastive learning view, this phrase grounding task only has two negatives. More negatives can be created from this annotation and thus enable stronger contrastive learning. In GLIPv2, we introduce the novel inter-image region-word contrastive learning task, which leverages phrases from other sentences in the same batch as potential negatives, as another much stronger VL grounding task. This new region-word contrastive loss enables GLIPv2 to learn more discriminative region-word features and demonstrates improvements over all downstream tasks.

**GLIPv2 achieves mutual benefit between localization and VL understanding**. 1) Experimental results (Table 2) show that a single GLIPv2 model (all model weights are shared) achieves near SoTA performance on various localization and understanding tasks. 2) Thanks to semantic-rich annotations from the image-text data, GLIPv2 shows superior zero-shot and few-shot transfer learning ability to open-world object detection and instance segmentation tasks, evaluated on the LVIS dataset and the "Object Detection in the Wild (ODinW)" benchmark. 3) GLIPv2 enables language-guided detection and segmentation ability, and achieves new SoTA performance on the Flick30K-entities phrase grounding and PhraseCut referring image segmentation tasks. 4) Inherently a grounding model, GLIPv2 leads to VL understanding models with strong grounding ability, which are self-explainable

and easy to debug. For example, GLIPv2, when GLIPv2 is finetuned on VQA, it can answer questions while localizing mentioned entities (see Figure 1 and Section 4.4).

## 2    Related Work

**Localization models.** Traditionally, localization tasks such as object detection and segmentation are single-modality and output bounding boxes or pixel masks [45, 38, 23, 14, 46, 10, 9]. One challenge of these single-modality models lies in generalization to rare and novel concepts: it is hard to collect localization data that cover many rare categories [20]. A long line of research focuses on this generalization problem, under the name of zero-shot [4, 62, 7, 63], weakly-supervised [18, 5, 52], or open-vocabulary [58, 19] localization. Built upon MDETR [25] and GLIP [36], GLIPv2 converts localization tasks into a grounded vision-language task using the classification-to-matching trick (Section 3). Thus GLIPv2 can learn from the semantic-rich vision-language data and shows strong performance on open-vocabulary localization tasks.

**Vision-language understanding models.** Vision-language (VL) understanding tasks such as VQA [3], image captioning [11], and image-text retrieval [26] involve understanding visual semantics and how they are expressed in natural language. Many VL models (e.g., BUTD) [2, 59] rely on a pre-trained localization model as their visual encoder; the downside is the pro-longed "localization->VLP" pre-training pipeline [41, 49, 13, 48, 34, 32, 61, 37, 35]. In contrast, GLIPv2 simplifies the pre-training pipeline and enables *grounded* VL understanding for better interpretability (Section 4.4).

**Unifying localization and understanding.** [21, 24, 56] made pioneering efforts in unifying localization and understanding. However, localization tasks are still treated as single-modality tasks, while VL tasks involve two modalities. The unification is achieved via straightforward multi-tasking: a low-level visual encoder is shared across tasks and two separate branches are designed for localization and VL understanding. Such unified models do not bring evident mutual benefit and often underperform task-specific models. In contrast, GLIPv2 identifies grounded VL understanding as a meta-task for localization and understanding. The task unification brings architecture unification: the unified grounded VL understanding model empowers a localization branch with VL capacity, arriving at a unified branch that excels at both tasks.

**GLIPv2 vs GLIP.** 1) GLIP shows that grounded pre-training improves localization. GLIPv2 further shows grounded pre-training improves VL understanding and thus leads to a unified model for localization and VL understanding. 2) GLIPv2 introduces the inter-image region-word contrastive loss, which is another and stronger grounding task than the pre-training task in GLIP. The proposed loss can be viewed as a region-word level generalization of the prevalent image-level contrastive learning [33, 44, 55]. 3) GLIPv2 outperforms GLIP on all benchmarks with the same pre-training data.

## 3    GLIPv2: Unifying Localization and VL Understanding

Based on the reformulation of object detection as a generalized phrase grounding task in GLIP [36], we unify both localization and VL understanding tasks as grounded vision-language tasks. A grounded vision-language task takes both image and text as inputs, and outputs region-level understanding results (e.g., detection, segmentation) and/or image-level understanding results with associated grounding/localization information (e.g., VQA, image captioning). We will present the unified grounded VL formulation and architecture in Section 3.1, the pre-training losses in Section 3.2, and transfer to downstream tasks in Section 3.3.

### 3.1    A Unified VL Formulation and Architecture

At the center of GLIPv2's unified formulation is the *classification-to-matching* trick, which reformulates any *task-specific fixed-vocab classification problem as an task-agnostic open-vocabulary vision-language matching* problem. The best example is the reformulation of image classification as image-text matching in CLIP [44], which enables the model to learn from raw image-text data directly, and achieves strong zero-shot results on open-vocabulary classification tasks. In GLIPv2, we

replace every semantic classification linear layer in traditional single-modality vision models with a vision-language matching dot-product layer.

As illustrated in Figure 1, GLIPv2's unified VL architecture is based on the generic architecture we term Architecture $\Pi$. It consists of a dual encoder, denoted as $\text{Enc}_V$ and $\text{Enc}_L$, and a fusion encoder, denoted as $\text{Enc}_{VL}$. The model takes an image-text pair (Img, Text) as input, and extract visual and text features as below:

$$\mathring{O} = \text{Enc}_V(\text{Img}), \quad \mathring{P} = \text{Enc}_L(\text{Text}), \quad O, P = \text{Enc}_{VL}(\mathring{O}, \mathring{P}), \tag{1}$$

where $(\mathring{O}, \mathring{P})$ and $(O, P)$ denote the image/text features *before* and *after* VL fusion, respectively.

**Vision-Language understanding tasks.** Arch $\Pi$ is the most popular model architecture for VL understanding tasks. Given the cross-modality fused representations $O$ and $P$, it is straightforward to add lightweight task-specific heads for various VL tasks. For example, GLIPv2 adds a two-layer MLP on top of text features $P$ as the masked language modeling (MLM) head, to perform the MLM pre-training. We provide model details of VQA and image captioning in Section 3.3.

**(Language-guided) object detection and phrase grounding.** Following GLIP [36], GLIPv2 uses the classification-to-matching trick to unify detection and grounding. More specifically, for detection, we simply replace the class logits $S_{\text{cls}} = OW^T$, where $W$ is the weight matrix of the box classifier, with a task-agnostic region-word similarity logits $S_{\text{ground}} = OP^T$, where text features $P$ are label embeddings from a task-agnostic language encoder. As shown in Figure 1, object detection and phrase grounding share the same input/output format and model architecture. See GLIP [36] for more details. Their only difference is the input text format: (1) for object detection, the text input is a string of concatenated candidate object labels; (2) for phrase grounding, the text input is a natural language sentence. We refer to GLIP [36] for more details.

**(Language-guided) instance segmentation and referring image segmentation.** Given the object detection results, an instance segmentation head is added to classify each pixel within the box into a semantic class. Again, GLIPv2 uses the classification-to-matching trick to produce a unified instance segmentation head for the standard instance segmentation tasks and the referring image segmentation tasks and leverage both types of data for its pre-training. This classification-to-matching trick can also apply to many other semantic classification heads in single modality CV models (e.g., semantic segmentation) and thus transfers them to language-guided CV models.

## 3.2 GLIPv2 Pre-training

The GLIPv2 is pre-trained with three pre-training losses: phrase grounding loss $\mathcal{L}_{\text{ground}}$ from a vision-language reformulation of the object detection task, region-word contrastive loss $\mathcal{L}_{\text{inter}}$ from a novel region-word level contrastive learning task, and the standard masked language modeling loss $\mathcal{L}_{\text{mlm}}$ proposed in BERT [16].

$$\mathcal{L}_{\text{GLIPv2}} = \underbrace{\mathcal{L}_{\text{loc}} + \mathcal{L}_{\text{intra}}}_{\mathcal{L}_{\text{ground}}} + \mathcal{L}_{\text{inter}} + \mathcal{L}_{\text{mlm}} \tag{2}$$

Similar to losses in detection tasks, the grounding loss $\mathcal{L}_{\text{ground}}$ has two parts: the localization loss $\mathcal{L}_{\text{loc}}$ trains localization heads with bounding-box supervision, e.g., RPN loss, box regression loss and/or centerness loss [50]; the intra-image region-word alignment loss $\mathcal{L}_{\text{intra}}$ is essentially the semantic classification/retrieval loss for each region.

**Intra-image region-word alignment loss.** Given one image-text pair (Img, Text), we obtain the image and text features *after* cross-modality fusion $O$ and $P$. The Intra-image region-word alignment loss is computed by

$$\mathcal{L}_{\text{intra}} = loss(OP^T; T), \tag{3}$$

where $OP^T$ is the similarity score between image regions and word tokens, and $T$ is the target affinity matrix determined by the ground-truth annotations. The loss function $loss$ is typically a cross-entropy loss for two-stage detectors [46] and a focal loss [38] for one-stage detectors.

However, as discussed in Section 1, this intra-image region-word contrastive learning is rather weak in the sense of contrastive learning, due to the limited number of phrases that can one caption can contain. GLIP [36] alleviates this problem by appending a few negative sentences to form a longer

text input with more (negative) phrases. However, constrained by the maximal length of text tokens (256 in GLIP and GLIPv2), only a few negative sentences can be added and the number of negative phrases remains in the order of 10's. This small-negative-example problem also exists in detection data [36] when the input text cannot include all class names in a detection dataset, e.g., Objects365.

**Inter-image region-word contrastive loss.** In GLIPv2, we propose using phrases from other image-text pairs in the same batch as negative examples, which effectively increases the number of negative examples to the order of 1000's, with nearly negligible additional computational cost.

As in (1), given a batch of image-text pairs $(\text{Img}^i, \text{Text}^i)_{i=1}^B$ and their ground-truth annotations $(T^i)_{i=1}^B$, the model produces the image and text features *before* and *after* VL fusion, denoted as $(\mathring{O}^i, \mathring{P}^i)_{i=1}^B$ and $(O^i, P^i)_{i=1}^B$, respectively. Then as illustrated in Figure 2 (Left), a batch-wise similarity matrix $S_{\text{ground}}^{\text{batch}}$ and a batch-wise target affinity matrix $T^{\text{batch}}$ are constructed by considering all the image regions and text phrases across this batch. Their $(i, j)$'th blocks are obtained as below:

$$S_{\text{ground}}^{\text{batch}}[i,j] = \mathring{O}^i(\mathring{P}^j)^T, \quad T^{\text{batch}}[i,j] = \begin{cases} T^i, & \text{if } i = j \\ \text{obtained by label propagation}, & \text{otherwise.} \end{cases} \quad (4)$$

The inter-image region-word contrastive loss is then defined as the standard bi-directional contrastive loss applied on all image regions and phrases in this batch:

$$\mathcal{L}_{\text{inter}} = \text{cross\_entropy\_loss}(S_{\text{ground}}^{\text{batch}}, T^{\text{batch}}, \text{axis} = 0) + \text{cross\_entropy\_loss}(S_{\text{ground}}^{\text{batch}}, T^{\text{batch}}, \text{axis} = 1). \quad (5)$$

Compared with that in the inter-image contrastive loss (3), the number of negatives is multiplied by batch size $B$ in this inter-image contrastive loss (5). We elaborate two important details in (4). (1) GLIPv2 uses the image text features $(\mathring{O}^i, \mathring{P}^i)_{i=1}^B$ before VL fusion, *not* $(O^i, P^i)_{i=1}^B$ after VL fusion, to compute the batch-wise similarity matrix in the inter-image contrastive loss (4). Otherwise, the image and text features after VL fusion would have seen the paired information (1), and thus the model can easily rule out the negatives from misaligned images/texts. (2) We cannot simply assign all regions and texts from unpaired image-text as negative pairs, as done in the standard contrastive loss in CLIP [44]. Instead, we determine the off-diagonal blocks in the target affinity matrix $T^{\text{batch}}$ by *label propagation*. For example, as illustrated in Figure 2 (Left), if a region is annotated as "person", it should be a positive pair with all "person" phrases in detection-type texts. We do not propagate positives to grounding-type texts (natural sentences) because phrases in sentences carry contexts that are unique to that image-sentence pair.

**Pre-training with both detection and paired-image-text data.** GLIPv2 pre-training data is in the image-text-target triplet format $(\text{Img}, \text{Text}, T)$, where the target affinity matrix $T$ contains the box-label localization annotations. We also use massive image-text pair data $(\text{Img}, \text{Text})$ to pre-train GLIPv2, by generating grounding boxes $\hat{T}$ for phrases in the text with the GLIP pre-trained model from [36]. The human-annotated OD/grounding data provides high-fidelity localization supervision, while the massive image-text data greatly improves the concept diversity for GLIPv2.

**Second-stage pre-training of the segmentation head.** GLIPv2 performs a second-stage pre-training of the language-guided segmentation head on both instance segmentation and image referring segmentation data, while fixing all other parts of the model.

### 3.3 Transfer GLIPv2 to Localization and VL Tasks

We introduce two ways to easily transfer GLIPv2 to various downstream tasks. In addition, GLIPv2 can perform conventional VL tasks (e.g., VQA) along with localization, effectively making every task we consider a "grounded VL understanding" task.

**One model architecture for all.** GLIPv2 can be transferred to downstream tasks by fine-tuning the model with an (optional) task-specific head. 1) For *detection and segmentation* tasks, no task-specific head is needed as the pre-training architecture can inherently perform detection and segmentation. 2) For *VL* tasks: for VQA, a classification head is added on top of the hidden representation of the start-of-sequence token; for caption generation, we train with a unidirectional language modeling loss, which maximizes the likelihood of the next word given context. We use a unidirectional attention mask and prevent the image part from attending to the text in the fusion layers.

**One set of weights for all.** There is a growing interest in developing models that can be transferred to various tasks while only changing the least amount of parameters to save training time and storage

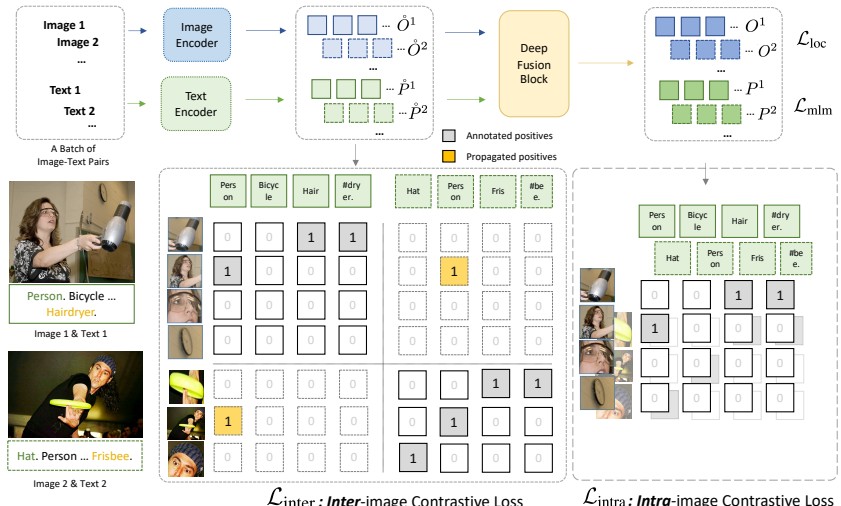

Figure 2: GLIPv2 pre-training losses: the intra-image alignment loss $\mathcal{L}_{\text{intra}}$ (right) takes features after VL fusion and compute loss over region-word pairs within each image-text pair; the inter-image contrastive loss (left) $\mathcal{L}_{\text{inter}}$ takes features before VL fusion and computes loss over all region-word pairs across a batch of image-text pairs. Label propagation is used to determine the off-diagonal blocks of the $\mathcal{L}_{\text{inter}}$ target matrix (4).

cost [47, 31]. Following GLIP, GLIPv2 can be transferred to localization tasks in a *zero-shot* or a *prompt-tuning* setting (Section 4.2). One single GLIPv2 model can serve various tasks, where each task only keeps few or no parameters. Of particular interest is the prompt tuning setting. For a certain localization task, the text prompt is the same for all input images; thus, we could directly tune $\mathring{P}$, a small prompt embedding matrix, to adapt GLIPv2 to new tasks. Prompt tuning in a deep-fused model such as GLIPv2 is different from the conventional linear probing/prompt tuning setting [53, 44, 60] in shallow-interacting vision models such as CLIP. The latter can also be viewed as only tuning a small prompt/softmax embedding $P$; however, tuning $P$ only affects the very last layer of the model while the visual representation is still frozen. In contrast, GLIP/GLIPv2's visual representation is conditioned on the prompt embedding $\mathring{P}$; tuning $\mathring{P}$ changes the text, visual, as well as fused embeddings. As a result, prompt tuning in GLIPv2 is highly effective, often matching the performance of fine-tuning (see Table 2). This is in contrast to the common observation in CV that linear probing lags behind fine-tuning by a large gap [22].

**Grounded VL understanding.** GLIPv2 also enables grounded VL understanding, where we retain the ability to perform grounding when fine-tuning the model to a downstream VL task. This increases the interpretability of the model. Specifically, we first turn the VL data of the downstream task into grounded VL data using a pre-trained GLIP model. Then we train the model with both the downstream task head and grounding head. For VQA, the model is trained to predict the answer and ground entities in the question as well as the implied entity in the answer; for captioning, the model is trained to predict the next word given the context and ground the current decoded word. By tuning localization tasks into a grounded VL task and augmenting VL tasks with grounding ability, we effectively turn every task into a grounded VL understanding task (see examples in Figure 1).

## 4   Experiments

In this section, we show that GLIPv2 serves as a performant and easy-to-deploy general-purpose vision system. 1) **One Model Architecture for All** (Section 4.1). GLIPv2 can be directly fine-tuned to both localization and VL understanding tasks with minimal architecture change. It achieves performance on par with SOTA models with specialized architectures. 2) **One Model Weight for All** (Section 4.2). GLIPv2 can be transferred to localization tasks in a zero-shot manner with zero parameter update; with prompt tuning, a single GLIPv2 model can achieve comparable performance with fully fine-tuned settings on both localization and understanding tasks.

| Model | Model Type | COCO-Det (test-dev) | ODinW (test) | LVIS (minival) | COCO-Mask (test-dev) | Flickr30K (test) | PhraseCut (test) | VQA (test-dev / test-std) | Captioning (Karpathy-test) |
|---|---|---|---|---|---|---|---|---|---|
| Mask R-CNN [23] | | 39.8 | - | 33.3 / - | - / 37.1 | - | - | - | - |
| DETR [9] | | 42.0 | - | 17.8 / - | - | - | - | - | - |
| DyHead-T [15] | Localization | 49.7 | 60.8 | - | - | - | - | - | - |
| DyHead-L [15] | | 60.3* | - | - | - | - | - | - | - |
| VisualBERT [34] | | - | - | - | - | 71.33 | - | 70.8 / 71.0 | - |
| UNITER [12] | Understanding | - | - | - | - | - | - | 73.8 / 74.0 | - |
| VinVL [59] | | - | - | - | - | - | - | **76.5 / 76.6** | 130.8 |
| GPV [21] | | - | - | - | - | - | - | 62.5 / - | 102.3 |
| UniT [24] | Localization & | 42.3 | - | - | - | - | - | 67.6 / - | - |
| MDETR [25] | Understanding | - | - | 24.2 / - | - | 84.3 | 53.7 | 70.6 / 70.6 | - |
| Unicorn [56] | | - | - | - | - | 80.4 | - | 69.2 / 69.4 | 119.1 |
| GLIP-T [36] | Localization & | 55.2 | 64.9 | - | - | 85.7 | - | - | - |
| GLIP-L [36] | Understanding | 61.5* | 68.9 | - | - | 87.1 | - | - | - |
| GLIPv2-T (**Ours**) | Localization | 55.5 | 66.5 | 50.6 / 41.4 | 53.5 / 42.0 | 86.5 | 59.4 | 71.6 / 71.8 | 122.1 |
| GLIPv2-B (**Ours**) | & | 58.8 | 69.4 | 57.3 / 46.2 | 59.0 / 45.8 | 87.5 | **61.3** | 73.1 / 73.3 | 128.5 |
| GLIPv2-H (**Ours**) | Understanding | **60.6 (62.4*)** | **70.4** | **59.8 / 48.8** | **59.8 / 48.9** | **87.7** | **61.3** | 74.6 / 74.8 | **131.0** |

Table 1: One model architecture results. For COCO-Det test-dev, * indicates multi-scale evaluation. For LVIS, we report the numbers for both `bbox` and `segm` on minival to avoid data contamination due to the pre-training. For Flickr30K test, we report the metric under `R@1`. For COCO-Mask, we also report both `bbox` and `segm` on test-dev.

Following GLIP [36], we adopt Swin Transformer [40] as the image encoder $Enc_V$, text transformers [51, 44] as the text encoder $Enc_L$, Dynamic Head [15] with language-aware deep fusion [36] as the fusion encoder $Enc_{VL}$, and Hourglass network [43] as instance segmentation head feature extractor. We train GLIPv2 at three scales: GLIPv2-T, GLIPv2-B, and GLIPv2-H.

**GLIPv2-T** has the same model config and initialization as GLIP-T: Swin-Tiny and BERT-Base as the dual encoder. The model is pre-trained on the following data: 1) O365, 2) GoldG as in GLIP-T (C), and 3) Cap4M, 4M image-text pairs collected from the web with boxes generated by GLIP-T [36]. **GLIPv2-B/GLIPv2-H** are based on Swin-Base/Swin-Huge and the pre-layernorm text transformer [17] as dual encoder, and are initialized from the UniCL [55] checkpoints. We observe much stabler training with GPT-type pre-layernorm transformer [17] than BERT-type post-layernorm transformer. The training data contain: 1) FiveODs (2.78M data) [1]; 2) GoldG as in MDETR [25]; and 3) CC15M+SBU, 16M public image-text data with generated boxes by GLIP-L [36]. **Segmentation heads** of GLIPv2 models are pre-trained on COCO, LVIS [20] and PhraseCut [54], with all other model parameters are frozen.

**Note** All datasets above were collected by the creators (cited) and consent for any personally identifiable information (PII) was ascertained by the authors where necessary. Due to limited space, we refer to supplementary for details of training recipes and hyper-parameters.

## 4.1 One Model Architecture for All

We compare GLIPv2 to existing object detection and vision-language pre-training methods on a wide range of tasks. We fine-tune the model on 8 different downstream tasks and report the performance in Table 1. We make the following observations.

**GLIPv2 v.s. specialized Localization methods.** GLIPv2 outperforms previous localization models on generalization to both common and rare classes and domains *with a single model architecture and pre-training stage*. *1) OD on common categories (COCO-Det)*, GLIPv2-T achieves 5.8 improvement compared to the standard DyHead-T trained on O365 (55.5 v.s. 49.7). GLIPv2-H reaches 62.4 AP on test-dev, and surpass the performance of the previous SoTA model GLIP-L. *2) OD on rare / unseen categories (LVIS)*, GLIPv2-T outperforms a supervised MDETR on the `bbox` by a great margin (59.8 v.s. 24.2). *3) Generalization to diverse real-word tasks (ODinw)*, GLIPv2-T (55.5) performs better than original GLIP-T (64.9) on the average of 13 public datasets; GLIPv2-B outperforms GLIP-L by 0.5 AP. *4) Instance segmentation (COCO-Mask & PhraseCut)*, for traditional instance segmentation (i.e., COCO-Mask), GLIPv2-H outperforms the well-known Mask R-CNN by a great margin on `segm`.

---

[1] Besides O365, it combines with 4 additional OD datasets including COCO [39], OpenImages [28], Visual Genome [29], and ImageNetBoxes [30]

| Model | Direct Evaluation | | | | Prompt Tuning | | | | |
|---|---|---|---|---|---|---|---|---|---|
| | COCO-Mask (minival) | ODinW (test) | LVIS-Det (minival) | Flickr30K (minival) | COCO-Det (test-dev) | ODinW (test) | LVIS (minival) | COCO-Mask (test-dev) | PhraseCut (test) |
| GLIP-T | 46.6/– | 46.5 | 26.0 | 85.7 | – | 46.5 | - | - | - |
| GLIP-L | 49.8/– | 52.1 | 37.3 | 87.1 | 58.8 | 67.9 | - | - | - |
| GLIPv2-T | **47.3**/35.7 | 48.5 | **29.0** | 86.0 | 53.4 (-2.1) | 64.8 (-1.7) | 49.3 / 34.8 (-1.3 / -6.6) | 53.2 / 41.2 (-0.3 / -0.8) | 49.4 |
| GLIPv2-B | 61.9†/43.4 | 54.2 | 48.5 | 87.2 | 59.0 (+0.2) | 67.3 (-2.1) | 56.8 / 41.7 (-0.5 / -4.5) | 58.8 / 44.9 (-0.2 / -0.9) | 55.9 |
| GLIPv2-H | 64.1†/47.4 | **55.5** | 50.1 | **87.7** | **60.2 / 61.9*** (-0.4 / -0.5) | **69.1** (-1.3) | **59.2 / 43.2** (-0.6 / -5.7) | **59.8 / 47.2** (-0.0 / -1.7) | **56.1** |

Table 2: One set of weights results v.s. Original GLIP. * indicates multi-scale evaluation. Numbers in red clearly points out the difference between the prompt tuning and full fine-tuning results (see Table 1). Numbers in gray mean that they are not in *zero-shot* manner. †: these two numbers are artificially high due to some overlap between COCO-minival and VisualGenome-train.

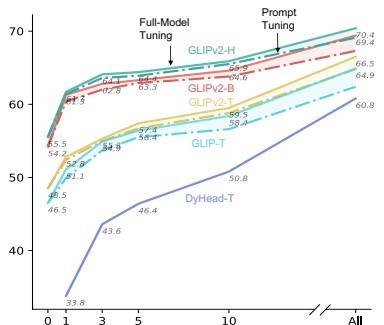

Figure 3: Data efficiency of GLIPv2 on ODinW. The X-axis is the amount of task-specific data, from zero-shot to all data. Y-axis is the average AP across 13 datasets.

| Model | Zero-Shot 0 | Prompt Tuning / Fine Tuning | | | | |
|---|---|---|---|---|---|---|
| | | 1 | 3 | 5 | 10 | All |
| DyHead-T O365 [36] | - | 33.8 | 43.6 | 46.4 | 50.8 | 60.8 |
| $\mathcal{L}_{loc} + \mathcal{L}_{intra}$ (GLIP-T) | 46.5 | 49.9 / 51.3 | 53.7 / 54.9 | 55.5 / 56.4 | 56.6 / 58.4 | 62.4 / 64.9 |
| $\mathcal{L}_{loc} + \mathcal{L}_{intra} + \mathcal{L}_{inter}$ | 48.4 | 52.1 / 51.4 | 55.6 / 55.3 | 56.7 / 56.6 | 58.3 / 59.5 | 62.9 / 66.3 |
| $\mathcal{L}_{loc} + \mathcal{L}_{intra} + \mathcal{L}_{inter} + \mathcal{L}_{mlm}$ | 48.5 | 52.4 / 52.8 | 55.6 / 55.6 | 57.4 / 57.4 | 58.8 / 59.7 | 64.8 / 66.5 |

Table 3: Zero-shot, prompt tuning, and full fine-tuning performance on ODinW. GLIPv2 models exhibit superior data efficiency.

For language-guided segmentation (i.e., PhraseCut), compared to MDETR, GLIPv2-T achieves an improvement of 5.7 mask AP.

**GLIPv2 v.s. specialized VL Understanding methods.** GLIPv2 rivals with SoTA specialized models for VL tasks. *1) For VQA*, GLIPv2 outperforms VisualBERT and UNITER and approaches the previous SoTA model VinVL. *2) For Captioning*, the best GLIPv2 even surpasses VinVL (VinVL and GLIPv2 are not trained with CIDEr optimization).

**GLIPv2 v.s. localization and VL models.** Prior works such GPV, UniT and Unicorn have also explored unifying localization and VL models (see a discussion in Section 2). GLIPv2 outperforms all previous systems on both localization and VL tasks. For the best GLIPv2-H, it outperforms the UniT by a great margin (18.3 AP) on COCO object detection tasks. Meanwhile, it also surpasses UniT's performance on VQA by 6.9 points and GPV's peformance on Image Captioning as well.

**Takeaway.** Most notably, GLIPv2 outperforms previous "unified" models (GPV, UniT, MDETR, Unicorn) by a large margin. This is the first time that a single model architecture could achieve near SoTA performance on both localization and understanding. In contrast, in prior work, there exists certain trade-off between localization and understanding: models that aim to achieve high understanding performance tend to have lower localization performance (e.g., UNiT's detection performance is limited to the DETR [9] architecture), as it is not trivial to merge a SoTA localization branch and a SoTA VL branch into a single model.

## 4.2 One Set of Model Parameters for All

GLIPv2 is pre-trained to perform grounding; thus it can be transferred to various localization tasks with changing zero or few parameters. We evaluate GLIPv2 under two such settings: 1) direct evaluation, where we transfer the model "as is" without any parameter change, and 2) prompt tuning, where only the prompt embedding is tuned for specific tasks (Section 3.3).

**Direct evaluation.** The pre-trained GLIPv2 can be directly evaluated on any object detection task (by concatenating the object categories into a text prompt) and visual grounding task without any further tuning. We evaluate the models on four localization tasks: COCO, ODinW, LVIS, and Flickr30, and their results are presented in Table 2. Note that for GLIPv2-B and GLIPv2-H, the training sets of Flick30K and LVIS are present in the pre-training data. Thus, reported numbers on these metrics are not *zero-shot* evaluation (we have marked them gray). For all other evaluation results, the models are evaluated in *zero-shot* settings without any further tuning.

*GLIPv2 can be effortlessly transferred to different localization tasks without further tuning.* 1) For *COCO*, GLIPv2-T achieves a zero-shot performance of 47.3 without seeing any COCO training images. This surpasses well-established supervised systems (e.g., Mask R-CNN) and also outperforms GLIP-T by 0.7 AP. 2) For *ODinW*, GLIPv2 also shows strong zero-shot performance. GLIPv2-T (48.5) surpasses the GLIP-T (46.5). Meanwhile, the zero-shot performance of GLIPv2-B and GLIPv2-H even surpasses the 10-shot tuning performance of DyHead-T (to be introduced in Figure 3). 3) For *LVIS*, GLIPv2-T achieves a 3 AP improvement performance compared to the GLIP-T. 4) For *Flickr30K*, GLIPv2-B achieves even higher number (87.2) compared to original GLIP-L (87.1).

**Prompt Tuning.** Following GLIP, GLIPv2 supports efficient prompt tuning: the visual representation is heavily conditioned on the text representation due to the deep fusion block (Section 3.3); thus we could fine-tune only the prompt embedding for each task but still maintain high performance.

*Prompt tuning GLIPv2 achieves similar performance as full fine-tuning.* When comparing the performance of each task in Table 1 and 2 at the same time, for GLIPv2, prompt tuning performance almost matches the one model architecture results on localization tasks, without changing any of the grounding model parameters.

### 4.3 GLIPv2 as a Strong Few-Shot Learner

We demonstrate GLIPv2's performance on ODinW datasets with respect to different amounts of training data in Figure 3. The performance improvement between GLIPv2-T and GLIP-T exhibits more superior data efficiency for prompt tuning. We compare with the SoTA detector DyHead-T, pre-trained on Objects365 in Table 3. It can be seen that a zero-shot GLIPv2-T (48.5) outperforms a outperforms 5-shot DyHead-T (46.4) while the performance of one-shot GLIPv2-H (61.3) surpasses a all-shot fully supervised DyHead-T (60.8).

### 4.4 Analysis

**Pre-training losses** Table 4 shows the performance of the downstream tasks with different variants of our method. Compared to the GLIP pre-training tasks with only intra-image region-word contrastive loss (Row 3), adding inter-image word-region loss (Row 5) substantially improves the pre-trained model performance across all the object detection tasks (COCO, ODinW, and LVIS) on both zero-shot and fine-tuned manner. Consistent with common observations from most VL understanding methods, adding MLM loss (Row4) benefits for learning the representation for understanding tasks (Flick30k, VQA, and Captioning). Furthermore, using all three losses together at the 1st stage pre-training and doing the 2nd stage pre-training without MLM on OD and GoldG data, GLIPv2 (Row6) can perform well on both the localization and VL understanding tasks.

An additional stage of pre-training is applied for small models (GLIPv2-T and GLIPv2-B) due to limited model capacity. In order to achieve higher performance on both localization and understanding tasks, we find that including all data (even with some noise) and MLM loss in the first stage of pre-training will benefit the model for learning a better representation of both localization and understanding capability. Since the OD tasks require the model with more accurate localization ability, in our 2nd stage of pre-training, we decide to eliminate the MLM loss. The large model (GLIPv2-H) does not need this additional stage because it has enough capacity to learn both word-region alignment and MLM together in a single stage.

**Pre-training data** Table 5 reports the last checkpoint results on GLIPv2 when we do the scaling up of pre-training data. As more weak image-text pair data (Cap) is involved in our training, it benefits both standard/in-domain (i.e., COCO, Flickr30K) and large-domain gap (i.e., ODinW, LVIS) tasks. We also show that by adding the inter-image region-word contrastive helps when we are fixing the data at the same scale. For large-domain gap tasks, adding the inter-image region-word contrastive

| Row | Model | COCO | ODinW | LVIS | Flickr30K | VQA | Captioning |
|-----|-------|------|-------|------|-----------|-----|------------|
| 1 | No pre-train | –/50.6 | –/60.8 | – | – | 64.6 | 111.5 |
| 2 | $+ \mathcal{L}_{\text{mlm}}$ | –/48.5 | –/37.4 | – | – | 64.6 | 110.9 |
| 3 | $+ \mathcal{L}_{\text{loc}} + \mathcal{L}_{\text{intra}}$ | 46.6/55.2 | 46.5/64.9 | 26.0 | 85.7 | 69.4 | 119.7 |
| 4 | $+ \mathcal{L}_{\text{loc}} + \mathcal{L}_{\text{intra}} + \mathcal{L}_{\text{mlm}}$ | 47.0/55.2 | 47.6/66.2 | 28.5 | 86.5 | 69.8 | 120.7 |
| 5 | $+ \mathcal{L}_{\text{loc}} + \mathcal{L}_{\text{intra}} + \mathcal{L}_{\text{inter}}$ | 47.1/55.4 | 48.4/66.3 | 28.6 | 85.8 | 68.7 | 120.4 |
| 6 | $+ \mathcal{L}_{\text{loc}} + \mathcal{L}_{\text{intra}} + \mathcal{L}_{\text{inter}} + \mathcal{L}_{\text{mlm}}$ | 47.3/55.5 | 48.5/66.5 | 29.0 | 86.3 | 70.7 | 122.1 |

Table 4: Pre-training losses on Tiny-scale model. Involving intra-image region-word alignment loss $\mathcal{L}_{\text{intra}}$, inter-image region-word contrastive loss $\mathcal{L}_{\text{inter}}$ and MLM loss $\mathcal{L}_{\text{mlm}}$ will benefit both localization and understanding tasks.

| $\mathcal{L}_{\text{inter}}$ | Pre-train Data | COCO | ODinW | LVIS | Flick30K |
|------|-------|------|-------|------|----------|
| ✗ | O365, GoldG | 48.06 | 43.14 | 25.6 | 84.36 |
| ✓ | O365, GoldG | 48.59 | 42.64 | 26.9 | 83.90 |
| ✗ | O365, GoldG, Cap4M | 48.21 | 51.35 | 34.2 | 85.56 |
| ✓ | O365, GoldG, Cap4M | 48.79 | 52.70 | 35.0 | 85.50 |
| ✗ | O365, GoldG, Cap12M | 48.50 | 49.32 | 35.5 | 85.79 |
| ✓ | O365, GoldG, Cap12M | 49.26 | 53.15 | 36.6 | 85.84 |

Table 5: Pre-train data scale up on Base-scale model. Results are reported at the last checkpoint. See supplementary for results at all checkpoints.

| Model | COCO Caption | | | Flickr30K Grounding | | |
|-------|------|-------|-------|------|------|------|
| | B4 | CIDEr | SPICE | R@1 | R@5 | R@10 |
| GLIPv2-T | 36.5 | 119.8 | 21.6 | 80.8 | 94.4 | 96.5 |
| GLIPv2-B | 37.4 | 123.0 | 21.9 | 81.0 | 94.5 | 96.5 |

Table 6: GLIPv2 can perform captioning and grounding at the same time (a.k.a., grounded VL understanding).

loss will further boost the model to learn better representation. For more detailed scaling-up effects on various tasks under all the checkpoints for GLIP and GLIPv2, refer to Appendix.

Note that the $(\text{Img}, \text{Text}, T)$ data used in GLIPv2 pre-training can be just human-annotated data (Row1&2 in Table 5), with which GLIPv2 pre-training does not involve any pseudo data from a pre-trained grounding/localization model. In order to achieve the best performance, GLIPv2 uses image-text pair data with pseudo boxes (Cap) from a pre-trained GLIP model (Row3-6 in Table 4), which is trained with the same "grounded VL understanding" task but just with smaller data.

**Grounded Vision-Language Understanding** GLIPv2 can be trained to perform a VL task and grounding at the same time (Section 3.3). We denote such an ability as grounded VL understanding. In Figure 1, we showcase grounded predictions of GLIPv2 on VQA and COCO captions. We also conduct quantitative evaluations (Table 6). The model achieves strong performance for both VL understanding (on COCO Caption) and localization (on Flickr30K Grounding). Such an ability to produce high-level semantic outputs (i.e., answers and captions) and supporting localization results is another appealing trait of GLIPv2, as potential users can have a better understanding of the model behaviour. See more detailed analysis and qualitative examples in the Appendix.

## 5 Conclusion and Social Impacts

This paper proposes GLIPv2, a unified framework for VL representation learning that serves both localization tasks and VL understanding tasks. We experimentally verify the effectiveness of the unified model and the novel region-word contrastive learning. Compared to existing methods, GLIPv2 achieves competitive near SoTA performance on various localization and understanding tasks. However, additional analysis of the data and the model is necessary before deploying it in practice since large-scale web data may contain unintended private information, unsuitable images/text, or some bias leakage. Further investigation may be needed for web data due to the above issues.

## 6 Acknowledgement

We thank anonymous reviewers for their comments and suggestions. Additional thanks go to the Microsoft Research Horizontal AI Team and Microsoft Alexander Multi-modal Team for providing computer resources for large-scale training. The baseline models used in our experiments are based on the open-source code released in the GitHub repository; we acknowledge all the authors who made their code public, which tremendously accelerates our project progress.

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
