# Appendix of GLIPv2: Unifying Localization and Vision-Language Understanding

**Haotian Zhang**[*1†], **Pengchuan Zhang**[*2†♠], **Xiaowei Hu**[3], **Yen-Chun Chen**[3], **Liunian Harold Li**[4†]
**Xiyang Dai**[3], **Lijuan Wang**[3], **Lu Yuan**[3], **Jenq-Neng Hwang**[1], **Jianfeng Gao**[3]
[1]University of Washington, [2]Meta AI, [3]Microsoft, [4]UCLA
{haotiz,hwang}@uw.edu,pengchuanzhang@fb.com,liunian.harold.li@cs.ucla.edu,
{Xiaowei.Hu,Yen-Chun.Chen,Xiyang.Dai,lijuanw,luyuan,jfgao}@microsoft.com

The appendix is organized as follows:

- In Section 1, we provide more visualizations of our model's predictions on various localization and VL understanding tasks.

- In Section 2, we describe all our evaluated tasks and their dataset in detail.

- In Section 3, we discuss the difference between our additional inter-image region-word contrastive loss and some other well-known losses that were also applied over a full batch in multiple works.

- In Section 4, we introduce the training details and hyperparameters used in Section 4 in the main paper.

- Section 5, we analyze the effect of using different language encoder and their pre-trained weights in our models.

- In Section 6, we provide more results for all the checkpoints of adding pre-training data (refer to Section 4 in the main paper).

- In Section 7, we provide a detailed analysis of the experiments of grounded captioning (mentioned in Section 4 in the main paper).

- In Section 8, we give out a comparison for the model's inference speed.

- In Section 9, we clearly provide the original sources of the images that are used in our paper.

- In Section 10, we present per-dataset results for all experiments in ODinW.

## 1 Visualization

We provide a clearer illustration of GLIPv2 in Figure 1, which elegantly unifies various localization (object detection, instance segmentation) and VL understanding (phrase grounding, VQA and captioning) tasks. More visualizations of the predictions under various tasks from GLIPv2 are also provided to indicate the model's strength and capability. Please refer to Figure 2 for OD / Grounding, Figure 3 for Instance / Referring Image Segmentation, and Figure 4 for Grounded VL Understanding.

## 2 Tasks and dataset descriptions

### 2.1 (Language-guided) object detection and phrase grounding

**COCO.** [1] The Microsoft Common Objects in Context dataset is a medium-scale object detection dataset. It has about 900k bounding box annotations for 80 object categories, with about 7.3 annotations per image. It is one of the most used object detection datasets, and its images are often used within other datasets (including VG and LVIS).

36th Conference on Neural Information Processing Systems (NeurIPS 2022).

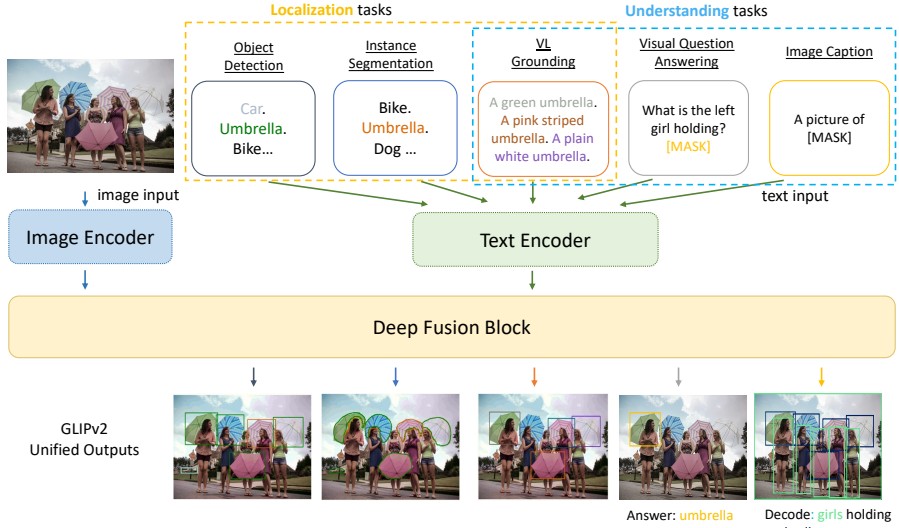

Figure 1: GLIPv2, a pre-trained grounded VL understanding model, unifies various localization and VL understanding tasks. These two kinds of tasks mutually benefit each other and enable new capabilities such as language-guided detection/segmentation and grounded VQA/captioning.

| Dataset | Objects of Interest | Train/Val/Test | URL |
|---|---|---|---|
| PascalVOC | Common objects (PascalVOC 2012) | 13690/3422/- | https://public.roboflow.com/object-detection/pascal-voc-2012 |
| AerialDrone | Boats, cars, etc. from drone images | 52/15/7 | https://public.roboflow.com/object-detection/aerial-maritime |
| Aquarium | Penguins, starfish, etc. in an aquarium | 448/127/63 | https://public.roboflow.com/object-detection/aquarium |
| Rabbits | Cottontail rabbits | 1980/19/10 | https://public.roboflow.com/object-detection/cottontail-rabbits-video-dataset |
| EgoHands | Hands in ego-centric images | 3840/480/480 | https://public.roboflow.com/object-detection/hands |
| Mushrooms | Two kinds of mushrooms | 41/5/5 | https://public.roboflow.com/object-detection/na-mushrooms |
| Packages | Delivery packages | 19/4/3 | https://public.roboflow.com/object-detection/packages-dataset |
| Raccoon | Raccoon | 150/29/17 | https://public.roboflow.com/object-detection/raccoon |
| Shellfish | Shrimp, lobster, and crab | 406/116/58 | https://public.roboflow.com/object-detection/shellfish-openimages |
| Vehicles | Car, bus, motorcycle, truck, and ambulance | 878/250/126 | https://public.roboflow.com/object-detection/vehicles-openimages |
| Pistols | Pistol | 2377/297/297 | https://public.roboflow.com/object-detection/pistols/1 |
| Pothole | Potholes on the road | 465/133/67 | https://public.roboflow.com/object-detection/pothole |
| Thermal | Dogs and people in thermal images | 142/41/20 | https://public.roboflow.com/object-detection/thermal-dogs-and-people |

Table 1: 13 ODinW dataset statistics. We summarize the objects of interest for each dataset and report the image number of each split.

**Flickr30k-entities.** [16] Given one or more phrases, which may be interrelated, the phrase grounding task is to provide a set of bounding boxes for each given phrase. We use the Flickr30k-entities dataset for this task, with the train/val/test splits as provided by [13] and evaluate our performance in terms of Recall. Flickr30K is included in the gold grounding data so we directly evaluate the models after pre-training as in MDETR [10]. We predict use any-box protocol specified in MDETR.

**ODinW.** We use 13 datasets from Roboflow[1]. Roboflow hosts over 30 datasets, and we exclude datasets that are too challenging (e.g., detecting different kinds of chess pieces) or impossible to solve without specific domain knowledge (e.g., understanding sign language). We provide the details of the 13 datasets we use in Table 1. We include the PASCAL VOC 2012 dataset as a reference dataset, as public baselines have been established on this dataset. For PascalVOC, we follow the convention and report on the validation set. For Pistols, there are no official validation or test sets so we split the dataset ourselves.

## 2.2 (Language-guided) instance segmentation and referring image segmentation

**LVIS.** [7] The Large Vocabulary Instance Segmentation dataset has over a thousand object categories, following a long-tail distribution with some categories having only a few examples. Similar to VG, LVIS uses the same images as in COCO, re-annotated with more object categories. In contrast to COCO, LVIS is a federated dataset, which means that only a subset of categories is annotated in each

---

[1]https://public.roboflow.com/object-detection

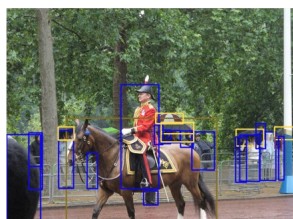 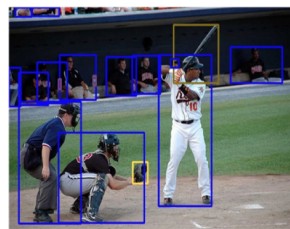 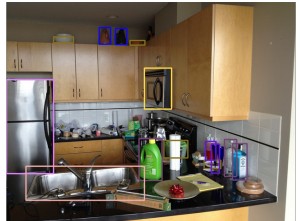

*Prompt:* person. dog ... backpack. umbrella. horse. toothbrush.

*Prompt:* person. hairdryer ... baseball bat. baseball glove. bottle. toothbrush.

*Prompt:* person. cup. sink ... microwave. refrigerator. bear.

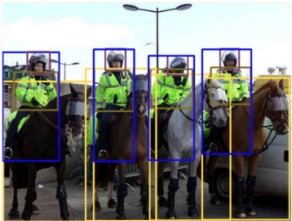 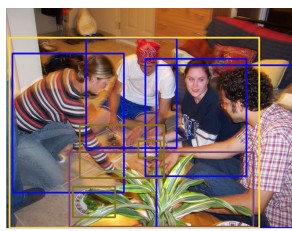 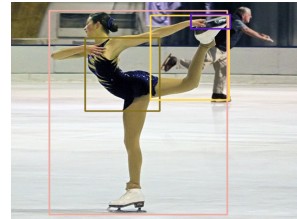

*Prompt:* Mounted officers in bright green jackets sit on their horses wearing helmets.

*Prompt:* 2 couples are eating dinner on the floor behind a large plant.

*Prompt:* A woman figure skater in a blue costume holds her leg by the blade of her skate

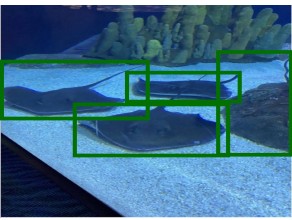 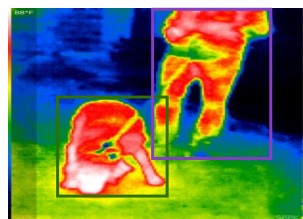 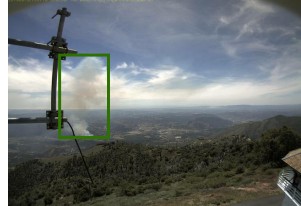

*Prompt:* fish. jellyfish. penguin. puffin. shark. starfish. stingray

*Prompt:* dog. person.

*Prompt:* smoke.

Figure 2: Visualization for OD / Grounding. Row 1: Object Detection on COCO. Row 2: Phrase Grounding on Flickr30K. Row 3: Object Detection on ODinW.

image. Annotations, therefore, include positive and negative object labels for objects that are present and categories that are not present, respectively. In addition, LVIS categories are not pairwise disjoint, such that the same object can belong to several categories.

**PhraseCut.** [19] Besides object detection, we show that our GLIPv2 can be extended to perform segmentation by evaluating the referring expression segmentation task of the recent PhraseCut[19] which consists of images from VG, annotated with segmentation masks for each referring expression. These expressions comprise a wide vocabulary of objects, attributes and relations, making it a challenging benchmark. Contrary to other referring expression segmentation datasets, in PhraseCut the expression may refer to several objects and the model is expected to find all the corresponding instances.

## 2.3 VQA and image captioning

**VQA.** [5] requires the model to predict an answer given an image and a question. We conduct experiments on the VQA2.0 dataset, which is constructed using images from COCO. It contains 83k images for training, 41k for validation, and 81k for testing. We treat VQA as a classification problem with an answer set of 3,129 candidates following the common practice of this task. For our best models, we report test-dev and test-std scores by submitting to the official evaluation server.[2]

---

[2] https://eval.ai/challenge/830/overview

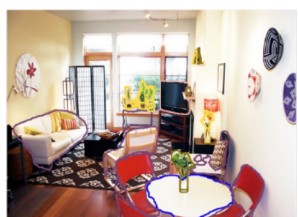 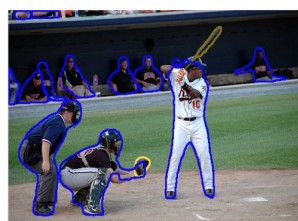 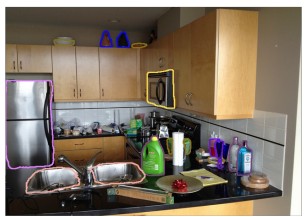

*Prompt:* person. chair. dining table … potted plant. vase.

*Prompt:* person. hairdryer ... baseball bat. baseball glove. bottle. toothbrush.

*Prompt:* person. cup. sink … microwave. refrigerator. bear.

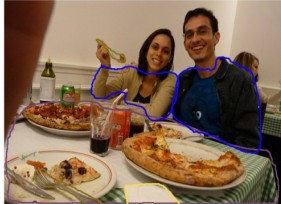 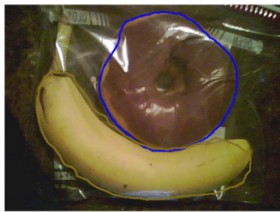 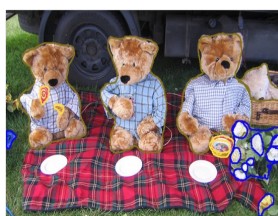

*Prompt:* tissue. jacket. … fork. pineapple. dinning table.

*Prompt:* donut. wineglass … banana. pineapple.

*Prompt:* person. teddy bear … lollipop. flower.

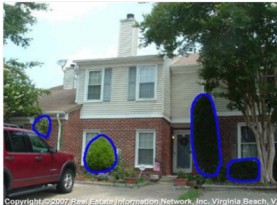 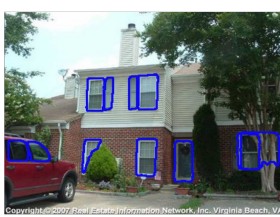 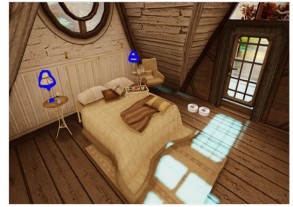

*Prompt:* green bush

*Prompt:* window has a frame

*Prompt:* brown lampshade

Figure 3: Visualization for Instance / Referring Image Segmentation. Row 1: Instance Segmentation on COCO Mask. Row 2: Instance Segmentation on LVIS. Row 3: Referring Image Segmentation on PhraseCut.

**COCO image captioning.** [2] The goal of image captioning is to generate a natural language description given an input image. We evaluate GLIPv2 on COCO Captioning dataset and report BLEU-4, CIDEr, and SPICE scores on the Karparthy test split.

## 3  Difference between inter-image region-word contrastive loss with other "region-word" losses.

As far as we know, up to the deadline (05/19/2022) for NeurIPS submission, there are only three published papers (VILD [6], RegionCLIP [24], and X-VLM [22]) that have the flavor of "region-word" loss applied over full batch. We discuss the difference between our work and the three aforementioned works in the following:

1. All these three works use "region-sentence" loss, i.e., the similarity between a region feature and the [CLS] token of a sentence, instead of true "region-word" loss used in GLIPv2. As a result, none of these three works made use of the phrase grounding data, which may contain multiple entities in one sentence during their training. It is the most important point in GLIPv2 to use phrase grounding data and pseudo grounding data to train a unified grounded VL understanding model.

2. GLIPv2 has carefully designed the positive label propagation in our inter-image region-word contrastive loss to mitigate the wrong assumption that "every unpaired region-word pair is negative". As far as we know, no previous work has mentioned this mechanism of positive label propagation before.

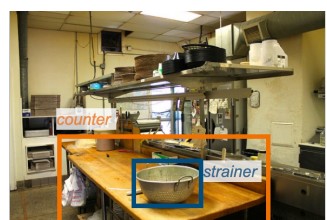 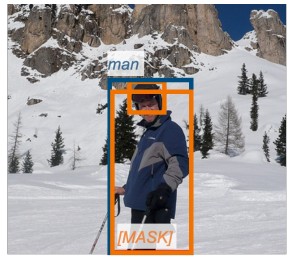 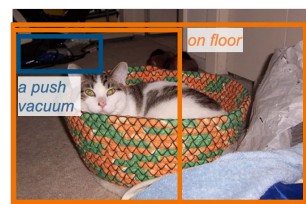

*Input:* Where is the strainer? [MASK]
*Prediction:* counter
*Gold:* counter

*Input:* What is the man wearing? [MASK]
*Prediction:* jacket
*Gold:* ski suit

*Input:* Where is a push vacuum? [MASK]
*Prediction:* on floor
*Gold:* background

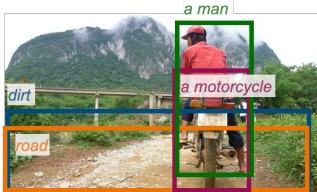 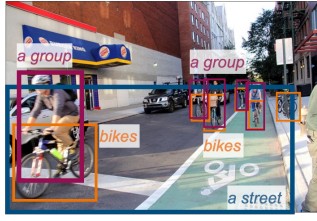 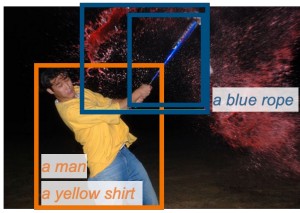

*Generated Caption:* a man riding a motorcycle on a dirt road.

*Generated Caption:* a group of people riding bikes down a street.

*Generated Caption:* a man in a yellow shirt is holding a blue rope.

Figure 4: Visualization for Grounded VL Understanding. Row 1: Grounded VQA predictions (The model is given the input question and a placeholder token "[MASK]" for the answer. The model can ground not only entities in the question but also the implied answer entity). Row 2: Grounded captioning on COCO (The model can generate high-quality captions and, in the meantime, provide localization results.

| Model | Image | Text | Pre-Train Data | | |
|---|---|---|---|---|---|
| | | | Detection | Grounding | Caption |
| GLIPv2-T | Swin-T | BERT-Base | O365 | GoldG (no COCO) | Cap4M |
| GLIPv2-B | Swin-B | CLIP | O365, COCO, OpenImages, VG, ImageNetBoxes | GoldG | CC15M+ SBU |
| GLIPv2-H | CoSwin-H [21] | CLIP | O365, COCO, OpenImages, VG, ImageNetBoxes | GoldG | CC15M+SBU |
| Mask Head | – | – | LVIS, COCO | PhraseCut | – |

Table 2: A detailed list of GLIPv2 model variants

3. There are some other differences. For example, in VILD, its "region-sentence loss" is actually not a contrastive loss over full-batch but a classification loss over a fixed vocabulary per sample (see the definition of $L_{ViLD-text}$).

Upon all three points above, we believe that our inter-image region-word contrastive loss is novel and has a significant difference from previous works.

## 4 Training details and hyperparamters

### 4.1 Pre-training

**Pre-training data.** There are three different types of data in pre-training 1) detection data 2) grounding data 3) caption data, as shown in Table 2. The detection data includes Object365 [17], COCO [1], OpenImages [11], Visual Genome [12], and ImageNetBoxes [3]. The grounding data includes GoldG, 0.8M human-annotated gold grounding data curated by MDETR [10] combining Flick30K, VG Caption, and GQA [9]. The Cap4M is a 4M image-text pairs collected from the web with boxes generated by GLIP-T(C) in [13], and CC (Conceptual Captions) + SBU (with 1M data).

**Implementation details.** In Section 4 in the main paper, we introduced GLIPv2-T, GLIPv2-B, GLIPv2-H, and we introduce the implementation details in the following.

We pre-train GLIPv2-T based on Swin-Tiny models with 32 GPUs and a batch size of 64. We use a base learning rate of $1 \times 10^{-5}$ for the language backbone (BERT-Base) and $1 \times 10^{-4}$ for all other parameters. The learning rate is stepped down by a factor of 0.1 at the 67% and 89% of the

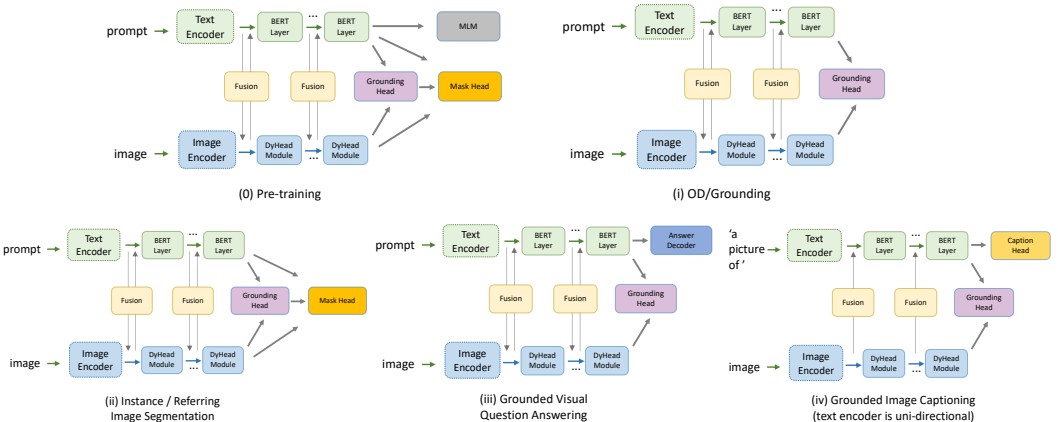

Figure 5: The model architecture for pre-training (0), and downstream tasks (i) OD / Grounding (ii) Instance / Referring Image Segmentation (iii) Grounded Visual Question Answering (iv) Grounded Image Captioning.

total 330,000 training steps. We decay the learning rate when the zero-shot performance on COCO saturates. The max input length is 256 tokens for all models. To optimize the results for object detection, we continue pre-training without the MLM loss for another 300,000 steps.

We pre-train GLIPv2-B based on Swin-Base models with 64 GPUs and a batch size of 64. We use a base learning rate of $1 \times 10^{-4}$ for all parameters, including the language backbone (CLIP-type pre-layernorm transformer). The learning rate is stepped down by a factor of 0.1 at the 67% and 89% of the total 1 million training steps. We decay the learning rate when the zero-shot performance on COCO saturates. The max input length is 256 tokens for all models. To optimize the results for object detection, we continue pre-training without the MLM loss for another 500,000 steps.

We pre-train GLIPv2-H based on the CoSwin-Huge model from Florence [21] with 64 GPUs and a batch size of 64. We use a base learning rate of $1 \times 10^{-4}$ for all parameters, including the language backbone (CLIP-type pre-layernorm transformer). The learning rate is stepped down by a factor of 0.1 at the 67% and 89% of the total 1 million training steps. We decay the learning rate when the zero-shot performance on COCO saturates. The max input length is 256 tokens for all models. We found that there is **no** need to continue pre-training without MLM loss for the huge model.

Mask heads of GLIPv2-T, GLIPv2-B and GLIPv2-H are pre-trained COCO, LVIS and PhraseCut, while freezing all the other model parameters. This mask head pre-training uses batch size 64, and goes through COCO for 24 epochs, LVIS for 24 epochs, and PhraseCut for 8 epochs, respectively. GLIPv2 uses Hourglass network [15] as instance segmentation head feature extractor, and utilizes the "classification-to-matching" trick to change the instance segmentation head linear prediction layer (outputs $K$-dimensional logits on each pixel) to a dot product layer between pixel visual features and the word features after VL fusion. GLIPv2-T and GLIPv2-B use a very basic Hourglass network for segmentation head feature extractor: only 1 scale and 1 layer, with hidden dimension 256. GLIPv2-H uses a larger Hourglass network for segmentation head feature extractor: 2 scales and 4 layers, with hidden dimension 384.

## 4.2 Downstream tasks

**OD / Grounding.** When fine-tuning on COCO, we use a base learning rate of $1 \times 10^{-5}$ and 24 training epochs for the pre-trained GLIPv2-T model, and a base learning rate of $5 \times 10^{-6}$ and 5 training epochs for the pre-trained GLIPv2-B and GLIPv2-H models.

For direct evaluation on LVIS, since LVIS has over 1,200 categories and they cannot be fit into one text prompt, so we segment them into multiple chunks, fitting 40 categories into one prompt and query the model multiple times with the different prompts. We find that models tend to overfit on LVIS during the course of pre-training so we closely monitor the performance on minival for all models and report the results with the best checkpoints in Table 2 in the main paper.

For direct evaluation on Flickr30K, models may also overfit during the course of pre-training so we monitor the performance on the validation set for all models and report the results with the best checkpoints in Table 2 in the main paper.

**Instance segmentation / Referring Image Segmentation.** Given the pre-trained model with pre-trained mask head, we simply fine-tune the **entire** network to get the task-specific fine-tuned models.

For fine-tuning on COCO instance segmentation, we use a base learning rate of $1 \times 10^{-5}$ and 24 training epochs for the pre-trained GLIPv2-T model, and a base learning rate of $5 \times 10^{-6}$ and 5 training epochs for the pre-trained GLIPv2-B and GLIPv2-H models.

For fine-tuning on LVIS instance segmentation, we use a base learning rate of $1 \times 10^{-5}$ and 24 training epochs for the pre-trained GLIPv2-T model, and a base learning rate of $5 \times 10^{-6}$ and 5 training epochs for the pre-trained GLIPv2-B and GLIPv2-H models.

For fine-tuning on PhraseCut Referring Image segmentation, we use a base learning rate of $1 \times 10^{-5}$ and 12 training epochs for the pre-trained GLIPv2-T model, and a base learning rate of $5 \times 10^{-6}$ and 3 training epochs for the pre-trained GLIPv2-B and GLIPv2-H models.

**(Grounded) VQA.** To fine-tune GLIPv2 for VQA, we feed the image and question into the model and then take the output feature sequence $P$ from the language side (after the VL fusion) and apply a 'attention pooling' layer to obtain a feature vector $P_{vqa}$. More specifically, the attention pooling layer applies a linear layer followed by softmax to obtain normalized scaler weights, and then these weights are used to compute a weighted sum to produce the feature vector $p_{vqa}$. This feature vector is then fed to a 2-layer MLP with GeLU activation [8] and a final linear layer to obtain the logits for the 3129-way classification.[3] Following standard practice [18], we use binary cross entropy loss to take account of different answers from multiple human annotators. Following VinVL [23], we train on the combination of train2014 + val2014 splits of the VQAv2 dataset, except for the reserved 2k dev split.[4] For the ablation studies we report the accuracy on this 2k dev split.

Other than the conventional VQA setting, we also experimented a new 'grounded VQA' setup, which the model is required to not only predict the answer, but also ground the objects (predict bounding boxes in the image) mentioned in the question and answer text, see Figure 5(iii). Note that the language input is the question appended by a [MASK] token, and this [MASK] token should ground to the object if the answer is indeed an object in the image. The total training loss is summing the grounding loss (intra-image region-word contrastive loss) and the VQA loss described previously.

**(Grounded) Image Captioning.** We fine-tune the pre-trained model on COCO Caption "Karpathy" training split. The training objective is uni-directional Language Modeling (LM), which maximizes the likelihood of the next word at each position given the image and the text sequence before it. To enable autoregressive generation, we use uni-directional attention mask for the text part, and prevent the image part from attending to the text part in the fusion layers. Although the training objective (LM) is different from that in pre-training (i.e., bi-directional MLM), we directly fine-tune the model for image captioning to evaluate its capability of generalizing to VL generation tasks. Our model is trained with cross entropy loss only, without using CIDEr optimization.

For grounded image captioning (Figure 5), we add the grounding loss (intra-image region-word contrastive loss) in training, which is calculated in the same way as in pre-training. We use Flickr30K training split for this task. During inference, for each predicted text token, we get its dot product logits with all the region representations and choose the maximum as the associated bounding box.

## 5 Analysis on the effect of different language encoders and pre-trained weights

For GLIPv2-T, we use the ImageNet pre-trained Swin-Transformer to initialize the image encoder and BERT-base-uncased to initialize the language encoder. For GLIPv2-B, we use the pre-trained paired image-language encoder from UniCL (CLIP-like pre-training, https://github.com/microsoft/UniCL) for initialization. We did an ablation study on the different language encoders (UniCL vs. BERT) and found that their results are nearly the same, as shown in Figure 6. Therefore, UniCL

---

[3] We experimented simpler pooling methods such as average pooling and [CLS] pooling [4] in the early experiments and found the attention pooling described above works better.

[4] 2000 images sampled from the val2014 split (and their corresponding question-answer pairs).

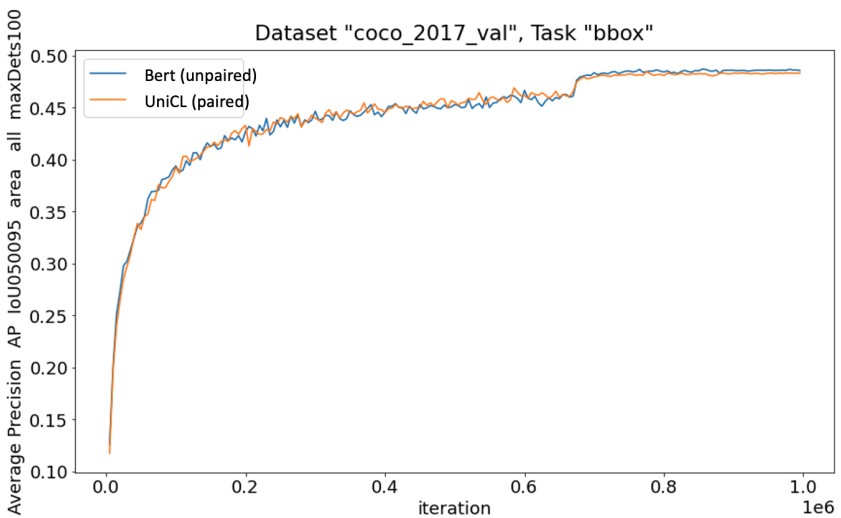

Figure 6: GLIP-B with image encoder initialized from UniCL pre-trained image encoder, but with different language encoder initialization. Blue: language encoder initialized by Bert-Base, thus un-paired image-language pre-trained encoders. Yellow: language encoder initialized from UniCL pre-trained language encoder, thus paired UniCL pre-trained image-language encoders. From the results, we can see that the COCO zero-shot transfer results from two initializations are nearly the same. Similar results have been observed for other metrics, i.e., LVIS zero-shot AP, ODinW benchmark, and Flickr30k grounding performance.

initialization does not skew the good localization performance. The main reason for us to keep the UniCL(CLIP-like) language encoder is due to its Pre-LayerNorm [20] operation. We find the UniCL(CLIP-like) language encoder with Pre-LayerNorm is more stable during the training compared with BERT, which uses Post-LayerNorm.

## 6  More analysis on pre-training data

Table 5 in the main paper reports the last checkpoint results on GLIPv2 when we do the scaling up of pre-training data. As more weak image-text pair data (Cap) is involved in our training, it benefits both standard/in-domain (i.e., COCO, Flickr30K) and large-domain gap (i.e., ODinW, LVIS) tasks. Further adding the inter-image region-word contrastive helps when we are fixing the data at the same scale. For large-domain gap tasks, adding the inter-image region-word contrastive loss will further boost the model to learn better representation. To learn more scaling-up effects on various tasks under all the checkpoints for GLIP and GLIPv2, see Figure 7. Given the considerable amount of improvement of GLIPv2 when the number of caption data increases from 0M to 12M, we hypothesize that it has potential to further grow by training on even larger-scale web image-text pairs.

## 7  Experiments on grounded image captioning

The grounded captioning task requires the model to generate an image caption and also ground predicted phrases to object regions. The final predictions consist of (1) the text captions (2) predicted object regions, and (3) the grounding correspondence between the phrases and regions. Following the established benchmarks [14, 25], we evaluate the caption metrics on COCO Captions and report the grounding metrics on Flick30K, as shown in Table 3.

## 8  Inference speed

We test the inference speed for GLIPv2 on V100 with batch size 1 and show its comparison to MDETR, as shown in Table 4.

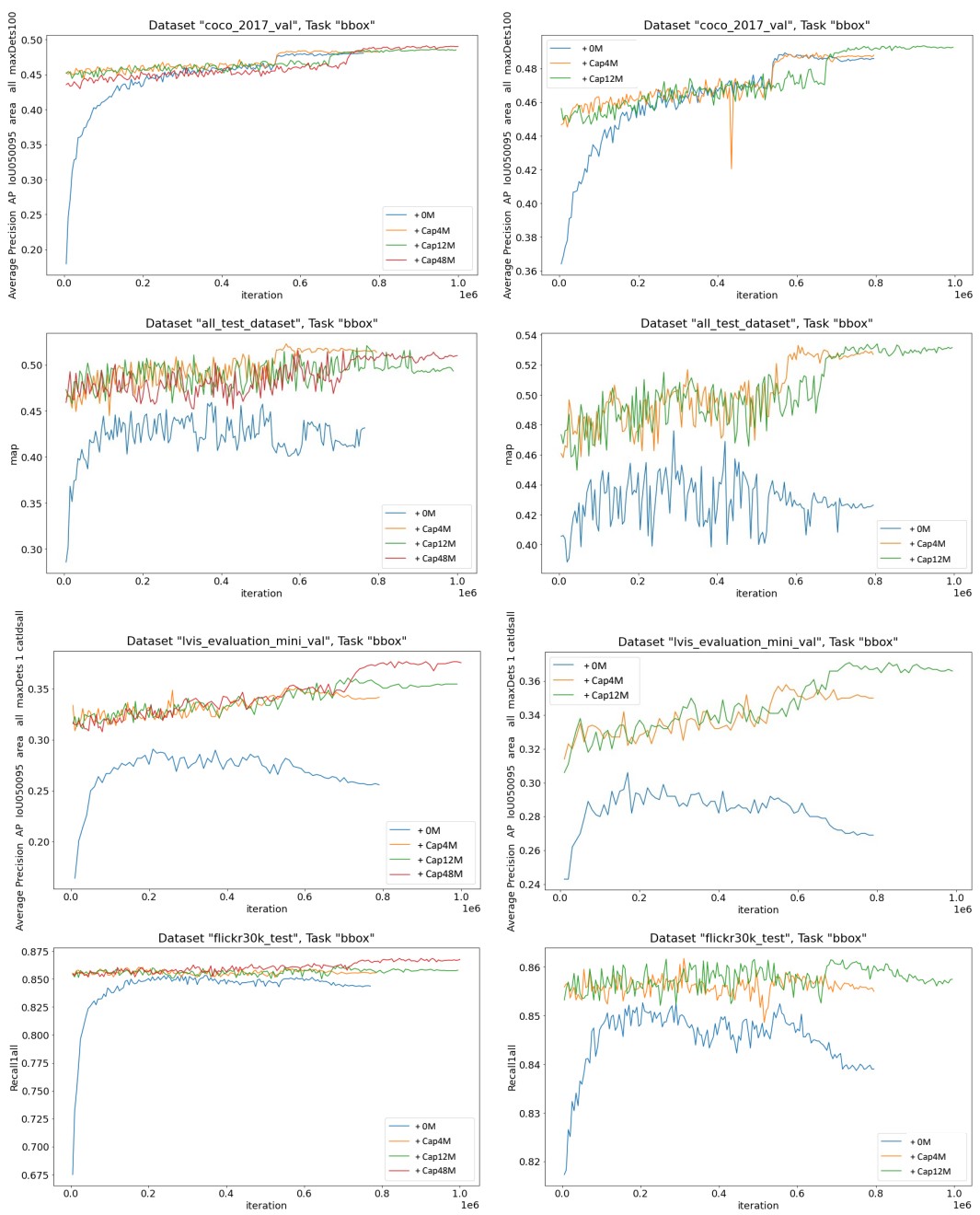

Figure 7: Pre-train data scale up on Base-scale model. Left: GLIP, Right: GLIPv2; Row 1: COCO minival, Row 2: ODinW test split, Row 3: LVIS minival, Row 4: Flick30K test.

# 9 Figure Reference

We provided the original sources of the images that are used in our paper in the following. All datasets above were collected by the creators (cited) and consent for any personally identifiable information (PII) was ascertained by the authors where necessary.

Figure 1 in the main paper - The top left and the bottom middle figures are the 281759.jpg in COCO val set; The left right images are (from top to down: (1) 2588.jpg in ODinW Aquarium test set. (2) 13923.jpg in LVIS val set. (3) 132690.jpg in VQA2.0 val set (question id is 132690002). (4) 462565.jpg in COCO Caption val set.

| Model | COCO Caption | | | Flickr30K Grounding | | |
| --- | --- | --- | --- | --- | --- | --- |
| | B@4 | CIDEr | SPICE | R@1 | R@5 | R@10 |
| No Pretrain | 35.4 | 115.3 | 21.2 | 77.0 | 92.9 | 95.7 |
| + $L_{\text{mlm}}$ | 33.4 | 107.6 | 19.9 | 70.9 | 90.0 | 93.2 |
| + $L_{\text{loc}} + L_{\text{intra}} + L_{\text{inter}}$ | 36.6 | 120.3 | 21.6 | 80.8 | 94.9 | 96.7 |
| GLIPv2-T | 36.5 | 119.8 | 21.6 | 80.8 | 94.4 | 96.5 |
| GLIPv2-B | 37.4 | 123.0 | 21.9 | 81.0 | 94.5 | 96.5 |

Table 3: Grounded image captioning results on the COCO Caption, and Flickr30K Entities. We report BLEU@4, CIDer, and SPICE metrics for caption evaluation, and we use R@1, R@5, R@10 for grounding evaluation.

| Model | Object Detection (COCO) | Phrase Grounding (Flick30K) | Referring Expression Segmentation (PhraseCut) |
| --- | --- | --- | --- |
| MDETR R101 [10] | – | 9.31 | 3.80 |
| MDETR EffB3 [10] | – | 11.20 | 3.98 |
| MDETR EffB5 [10] | – | 9.15 | – |
| GLIPv2-T | 4.12 | 3.74 | 2.26 |
| GLIPv2-B | 3.01 | 3.23 | 2.39 |
| GLIPv2-H | 1.21 | 1.13 | 0.89 |

Table 4: Model inference speed on various tasks. We report FPS, which is the number of images processed per second per GPU (higher is better).

Figure 2 in the main paper - The top left figure is the 209297.jpg in COCO train set; The bottom left figure is the 9378.jpg in COCO val set.

Figure 1 in the Appendix - Same as Figure 1 in the main paper. The top left and the bottom middle figures are the 281759.jpg in COCO val set.

Figure 2 in the Appendix - Row 1 (from left to right): (1) 439715.jpg in COCO val set. (2) 6471.jpg in COCO val set. (3) 13923.jpg in COCO val set; Row 2: (1) 5521996.jpg in Flickr30K val set. (2) 764507.jpg in Flickr30K val set. (3) 7520721.jpg in Flick30K val set; Row 3: (1) 2588.jpg in ODinW Aquarium test set. (2) 143.jpg in Thermal val set. (3) ck0l9j6n6oqjo0848ps5blk3b.jpg in WildFire val set.

Figure 3 in the Appendix - Row 1 (from left to right): (1) 13923.jpg in COCO val set. (2) 6471.jpg in COCO val set. (3) 7574.jpg in COCO val set; Row 2: (1) 117320.jpg in LVIS val set. (2) 2587.jpg in LVIS val set. (3) 211120.jpg in LVIS val set; Row 3: (1) 4744.jpg in PhraseCut test set. (2) 4744.jpg in PhraseCut val set. (3) 567.jpg in PhraseCut train set.

Figure 4 in the Appendix - Row 1 (from left to right): (1) 486.jpg in VQA2.0 val set (question id is 486002). (2) 262746.jpg in VQA2.0 val set (question id is 262746002). (3) 132690.jpg in VQA2.0 val set (question id is 132690002); Row 2: (1) 391895.jpg in COCO Caption val set. (2) 462565.jpg in COCO Caption val set. (3) 579056.jpg in COCO Caption val set.

## 10 All results for ODinW

We report the per-dataset performance under 0,1,3,5,10-shot and full data as well as prompt tuning, and full-model tuning in Table 5 and Table 6 (on the next page).

| Model | PascalVOC | AerialDrone | Aquarium | Rabbits | EgoHands | Mushrooms | Packages | Raccoon | Shellfish | Vehicles | Pistols | Pothole | Thermal | Avg |
| --- | --- | --- | --- | --- | --- | --- | --- | --- | --- | --- | --- | --- | --- | --- |
| GLIP-T | 56.2 | 12.5 | 18.4 | 70.2 | 50.0 | 73.8 | 72.3 | 57.8 | 26.3 | 56.0 | 49.6 | 17.7 | 44.1 | 46.5 |
| GLIP-L | 61.7 | 7.1 | 26.9 | 75.0 | 45.5 | 49.0 | 62.8 | 63.3 | 68.9 | 57.3 | 68.6 | 25.7 | 66.0 | 52.1 |
| GLIPv2-T | 57.6 | 10.5 | 18.4 | 71.4 | 52.7 | 77.7 | 67.7 | 58.8 | 27.8 | 55.6 | 60.1 | 20.0 | 52.4 | 48.5 |
| GLIPv2-B | 62.8 | 8.6 | 18.9 | 73.7 | 50.3 | 83.0 | 68.6 | 61.6 | 56.0 | 53.8 | 67.8 | 32.6 | 53.8 | 54.2 |
| GLIPv2-H | 66.3 | 10.9 | 30.4 | 74.6 | 55.1 | 52.1 | 71.3 | 63.8 | 66.2 | 57.2 | 66.4 | 33.8 | 73.3 | 55.5 |

Table 5: Zero-shot performance on 13 ODinW datasets.

| Model | Shot | Tune | PascalVOC | AerialDrone | Aquarium | Rabbits | EgoHands | Mushrooms | Packages | Raccoon | Shellfish | Vehicles | Pistols | Pothole | Thermal | Avg |
|---|---|---|---|---|---|---|---|---|---|---|---|---|---|---|---|---|
| DyHead $_{O365}$ | 1 | Full | $25.8_{\pm3.0}$ | $16.5_{\pm1.8}$ | $15.9_{\pm2.7}$ | $55.7_{\pm6.0}$ | $44.0_{\pm3.6}$ | $66.9_{\pm3.9}$ | $54.2_{\pm5.7}$ | $50.7_{\pm7.7}$ | $14.1_{\pm3.6}$ | $33.0_{\pm11.0}$ | $11.0_{\pm6.5}$ | $8.2_{\pm4.1}$ | $43.2_{\pm10.0}$ | $33.8_{\pm3.5}$ |
| DyHead $_{O365}$ | 3 | Full | $40.4_{\pm1.0}$ | $20.5_{\pm4.0}$ | $26.5_{\pm1.3}$ | $57.9_{\pm2.0}$ | $53.9_{\pm2.5}$ | $76.5_{\pm2.3}$ | $62.6_{\pm13.3}$ | $52.5_{\pm5.0}$ | $22.4_{\pm1.7}$ | $47.4_{\pm2.0}$ | $30.1_{\pm6.9}$ | $19.7_{\pm1.5}$ | $57.0_{\pm2.3}$ | $43.6_{\pm1.0}$ |
| DyHead $_{O365}$ | 5 | Full | $43.5_{\pm1.0}$ | $25.3_{\pm1.8}$ | $35.8_{\pm0.5}$ | $63.0_{\pm1.0}$ | $56.2_{\pm3.9}$ | $76.8_{\pm5.9}$ | $62.5_{\pm8.7}$ | $46.6_{\pm3.1}$ | $28.8_{\pm2.2}$ | $51.2_{\pm2.2}$ | $38.7_{\pm4.1}$ | $21.0_{\pm1.4}$ | $53.4_{\pm5.2}$ | $46.4_{\pm1.1}$ |
| DyHead $_{O365}$ | 10 | Full | $46.6_{\pm1.0}$ | $29.0_{\pm2.8}$ | $41.7_{\pm1.0}$ | $65.2_{\pm2.5}$ | $62.5_{\pm0.8}$ | $85.4_{\pm2.2}$ | $67.9_{\pm4.5}$ | $47.9_{\pm2.2}$ | $28.6_{\pm5.0}$ | $53.8_{\pm1.0}$ | $39.2_{\pm4.9}$ | $27.9_{\pm2.3}$ | $64.1_{\pm2.6}$ | $50.8_{\pm1.3}$ |
| DyHead $_{O365}$ | All | Full | 53.3 | 28.4 | 49.5 | 73.5 | 77.9 | 84.0 | 69.2 | 56.2 | 43.6 | 59.2 | 68.9 | 53.7 | 73.7 | 60.8 |
| GLIP-T | 1 | Prompt | $54.4_{\pm0.9}$ | $15.2_{\pm1.4}$ | $32.5_{\pm1.0}$ | $68.0_{\pm3.2}$ | $60.0_{\pm0.7}$ | $75.8_{\pm1.2}$ | $72.3_{\pm0.0}$ | $54.5_{\pm3.9}$ | $24.1_{\pm1.30}$ | $59.2_{\pm0.9}$ | $57.4_{\pm0.6}$ | $18.9_{\pm1.8}$ | $56.9_{\pm2.7}$ | $49.9_{\pm0.6}$ |
| GLIP-T | 3 | Prompt | $56.8_{\pm0.8}$ | $18.9_{\pm3.6}$ | $37.6_{\pm1.6}$ | $72.4_{\pm0.5}$ | $62.8_{\pm1.3}$ | $85.4_{\pm2.8}$ | $64.5_{\pm4.6}$ | $69.1_{\pm1.8}$ | $22.0_{\pm0.9}$ | $62.7_{\pm1.1}$ | $56.1_{\pm0.6}$ | $25.9_{\pm0.7}$ | $63.8_{\pm4.8}$ | $53.7_{\pm1.3}$ |
| GLIP-T | 5 | Prompt | $58.5_{\pm0.5}$ | $18.2_{\pm0.1}$ | $41.0_{\pm1.2}$ | $71.8_{\pm2.4}$ | $65.7_{\pm0.7}$ | $87.5_{\pm2.2}$ | $72.3_{\pm0.0}$ | $60.6_{\pm2.2}$ | $31.4_{\pm4.2}$ | $61.0_{\pm1.8}$ | $54.4_{\pm0.6}$ | $32.6_{\pm1.4}$ | $66.3_{\pm2.8}$ | $55.5_{\pm0.5}$ |
| GLIP-T | 10 | Prompt | $59.7_{\pm0.7}$ | $19.8_{\pm1.6}$ | $44.8_{\pm0.9}$ | $72.1_{\pm1.0}$ | $65.9_{\pm0.6}$ | $87.4_{\pm1.1}$ | $72.3_{\pm0.0}$ | $57.5_{\pm1.2}$ | $30.0_{\pm1.4}$ | $62.1_{\pm1.4}$ | $57.8_{\pm0.9}$ | $33.5_{\pm0.1}$ | $73.1_{\pm1.4}$ | $56.6_{\pm0.2}$ |
| GLIP-T | All | Prompt | 66.4 | 27.6 | 50.9 | 70.6 | 73.3 | 88.1 | 75.6 | 64.0 | 40.3 | 65.4 | 68.3 | 50.7 | 78.5 | 62.4 |
| GLIP-T | 1 | Full | $54.8_{\pm2.0}$ | $18.4_{\pm1.0}$ | $33.8_{\pm1.1}$ | $70.1_{\pm2.9}$ | $64.2_{\pm1.8}$ | $83.7_{\pm3.0}$ | $70.8_{\pm2.1}$ | $56.2_{\pm1.8}$ | $22.9_{\pm0.2}$ | $56.6_{\pm0.5}$ | $59.9_{\pm0.4}$ | $18.9_{\pm3.4}$ | $54.5_{\pm2.7}$ | $51.1_{\pm0.1}$ |
| GLIP-T | 3 | Full | $58.1_{\pm0.5}$ | $22.9_{\pm1.3}$ | $40.8_{\pm0.9}$ | $65.7_{\pm1.6}$ | $66.0_{\pm0.2}$ | $84.7_{\pm0.5}$ | $65.7_{\pm2.8}$ | $62.6_{\pm1.4}$ | $27.2_{\pm2.7}$ | $61.9_{\pm1.8}$ | $60.7_{\pm0.2}$ | $27.1_{\pm1.2}$ | $70.4_{\pm2.5}$ | $54.9_{\pm0.2}$ |
| GLIP-T | 5 | Full | $59.5_{\pm0.4}$ | $23.8_{\pm0.9}$ | $43.6_{\pm1.4}$ | $68.7_{\pm1.3}$ | $66.1_{\pm1.6}$ | $85.4_{\pm0.4}$ | $72.3_{\pm0.0}$ | $62.1_{\pm1.2}$ | $27.3_{\pm1.2}$ | $61.0_{\pm1.8}$ | $62.7_{\pm1.6}$ | $34.5_{\pm0.5}$ | $66.6_{\pm2.3}$ | $56.4_{\pm0.4}$ |
| GLIP-T | 10 | Full | $59.1_{\pm1.3}$ | $26.3_{\pm1.1}$ | $46.3_{\pm1.6}$ | $67.3_{\pm1.5}$ | $67.1_{\pm0.7}$ | $87.8_{\pm0.5}$ | $72.3_{\pm0.0}$ | $57.7_{\pm1.7}$ | $34.6_{\pm1.7}$ | $65.4_{\pm1.4}$ | $61.6_{\pm1.0}$ | $39.3_{\pm1.0}$ | $74.7_{\pm2.3}$ | $58.4_{\pm0.2}$ |
| GLIP-T | All | Full | 62.3 | 31.2 | 52.5 | 70.8 | 78.7 | 88.1 | 75.6 | 61.4 | 51.4 | 65.3 | 71.2 | 58.7 | 76.7 | 64.9 |
| GLIP-L | 1 | Prompt | $62.8_{\pm0.4}$ | $18.0_{\pm1.8}$ | $37.4_{\pm0.3}$ | $71.9_{\pm2.4}$ | $68.9_{\pm0.1}$ | $81.8_{\pm3.4}$ | $65.0_{\pm2.8}$ | $63.9_{\pm0.4}$ | $70.2_{\pm1.2}$ | $67.0_{\pm0.4}$ | $69.3_{\pm0.1}$ | $27.6_{\pm0.4}$ | $69.8_{\pm0.6}$ | $59.5_{\pm0.4}$ |
| GLIP-L | 3 | Prompt | $65.0_{\pm0.5}$ | $21.4_{\pm1.0}$ | $43.6_{\pm1.1}$ | $72.9_{\pm0.7}$ | $70.4_{\pm0.1}$ | $91.4_{\pm0.7}$ | $57.7_{\pm3.7}$ | $70.7_{\pm1.1}$ | $69.7_{\pm0.9}$ | $62.6_{\pm0.8}$ | $67.7_{\pm0.4}$ | $36.2_{\pm1.1}$ | $68.8_{\pm1.5}$ | $61.4_{\pm0.3}$ |
| GLIP-L | 5 | Prompt | $65.6_{\pm0.3}$ | $19.9_{\pm1.6}$ | $47.7_{\pm0.7}$ | $73.7_{\pm0.7}$ | $70.6_{\pm0.3}$ | $86.8_{\pm0.5}$ | $64.6_{\pm0.7}$ | $69.4_{\pm3.3}$ | $68.0_{\pm1.3}$ | $67.8_{\pm1.5}$ | $68.3_{\pm0.3}$ | $36.6_{\pm1.6}$ | $71.9_{\pm0.6}$ | $62.4_{\pm0.5}$ |
| GLIP-L | 10 | Prompt | $65.9_{\pm0.2}$ | $23.4_{\pm2.6}$ | $50.3_{\pm0.7}$ | $73.6_{\pm0.7}$ | $71.8_{\pm0.3}$ | $86.5_{\pm0.3}$ | $70.5_{\pm1.1}$ | $69.0_{\pm0.5}$ | $69.4_{\pm2.4}$ | $70.8_{\pm1.2}$ | $68.8_{\pm0.6}$ | $39.3_{\pm0.9}$ | $74.9_{\pm2.1}$ | $64.2_{\pm0.4}$ |
| GLIP-L | All | Prompt | 72.9 | 23.0 | 51.8 | 72.0 | 75.8 | 88.1 | 75.2 | 69.5 | 73.6 | 72.1 | 73.7 | 53.5 | 81.4 | $67.9_{\pm0.0}$ |
| GLIP-L | 1 | Full | $64.8_{\pm0.6}$ | $18.7_{\pm0.6}$ | $39.5_{\pm1.2}$ | $70.0_{\pm1.5}$ | $70.5_{\pm0.2}$ | $69.8_{\pm18.0}$ | $70.6_{\pm4.0}$ | $68.4_{\pm1.2}$ | $71.0_{\pm1.3}$ | $65.4_{\pm1.1}$ | $68.1_{\pm2.0}$ | $28.9_{\pm2.9}$ | $72.9_{\pm4.7}$ | $59.9_{\pm1.4}$ |
| GLIP-L | 3 | Full | $65.6_{\pm0.6}$ | $22.3_{\pm1.1}$ | $45.2_{\pm0.4}$ | $72.3_{\pm1.4}$ | $70.4_{\pm0.4}$ | $81.6_{\pm13.3}$ | $71.8_{\pm0.3}$ | $65.3_{\pm1.6}$ | $67.6_{\pm1.0}$ | $66.7_{\pm0.9}$ | $68.1_{\pm0.3}$ | $37.0_{\pm1.9}$ | $73.1_{\pm3.3}$ | $62.1_{\pm0.7}$ |
| GLIP-L | 5 | Full | $66.0_{\pm0.4}$ | $26.4_{\pm2.5}$ | $49.5_{\pm1.1}$ | $70.7_{\pm0.2}$ | $71.9_{\pm0.2}$ | $88.1_{\pm0.0}$ | $71.1_{\pm0.6}$ | $68.8_{\pm1.2}$ | $68.5_{\pm1.7}$ | $70.0_{\pm0.9}$ | $68.3_{\pm0.5}$ | $39.9_{\pm1.4}$ | $75.2_{\pm2.7}$ | $64.2_{\pm0.3}$ |
| GLIP-L | 10 | Full | $66.4_{\pm0.7}$ | $32.0_{\pm1.4}$ | $52.3_{\pm1.1}$ | $70.6_{\pm0.7}$ | $72.4_{\pm0.3}$ | $88.1_{\pm0.0}$ | $67.1_{\pm3.6}$ | $64.7_{\pm3.1}$ | $69.4_{\pm1.4}$ | $71.5_{\pm0.8}$ | $68.4_{\pm0.7}$ | $44.3_{\pm0.6}$ | $76.3_{\pm1.1}$ | $64.9_{\pm0.7}$ |
| GLIP-L | All | Full | 69.6 | 32.6 | 56.6 | 76.4 | 79.4 | 88.1 | 67.1 | 69.4 | 65.8 | 71.6 | 75.7 | 60.3 | 83.1 | 68.9 |
| GLIPv2-T | 1 | Prompt | $51.2_{\pm0.9}$ | $17.7_{\pm1.2}$ | $34.2_{\pm0.1}$ | $68.7_{\pm1.2}$ | $67.3_{\pm0.9}$ | $83.7_{\pm2.1}$ | $68.1_{\pm1.7}$ | $53.4_{\pm0.2}$ | $30.0_{\pm0.9}$ | $59.0_{\pm0.0}$ | $60.0_{\pm0.7}$ | $21.9_{\pm0.6}$ | $66.5_{\pm0.7}$ | $52.4_{\pm0.5}$ |
| GLIPv2-T | 3 | Prompt | $66.6_{\pm0.2}$ | $11.5_{\pm0.7}$ | $37.2_{\pm1.0}$ | $71.7_{\pm0.3}$ | $70.1_{\pm0.4}$ | $45.7_{\pm0.1}$ | $57.7_{\pm1.2}$ | $69.7_{\pm1.5}$ | $42.7_{\pm0.4}$ | $67.5_{\pm0.9}$ | $65.6_{\pm1.0}$ | $36.7_{\pm1.2}$ | $69.2_{\pm1.2}$ | $55.6_{\pm0.4}$ |
| GLIPv2-T | 5 | Prompt | $58.9_{\pm1.2}$ | $17.4_{\pm0.6}$ | $42.8_{\pm0.4}$ | $72.6_{\pm0.5}$ | $66.1_{\pm0.2}$ | $84.9_{\pm0.8}$ | $69.7_{\pm0.6}$ | $65.5_{\pm2.1}$ | $35.6_{\pm0.8}$ | $62.8_{\pm0.9}$ | $59.8_{\pm0.2}$ | $35.5_{\pm0.9}$ | $74.4_{\pm0.2}$ | $57.4_{\pm0.4}$ |
| GLIPv2-T | 10 | Prompt | $59.9_{\pm0.4}$ | $21.6_{\pm2.0}$ | $43.7_{\pm0.3}$ | $74.3_{\pm0.4}$ | $68.2_{\pm0.7}$ | $88.1_{\pm0.1}$ | $72.0_{\pm0.9}$ | $60.0_{\pm0.4}$ | $35.6_{\pm1.2}$ | $66.1_{\pm0.6}$ | $61.0_{\pm0.3}$ | $42.8_{\pm0.4}$ | $70.9_{\pm3.2}$ | $58.8_{\pm0.5}$ |
| GLIPv2-T | All | Prompt | 67.4 | 22.3 | 50.5 | 74.3 | 73.4 | 85.5 | 74.7 | 65.8 | 53.7 | 67.4 | 68.9 | 52.3 | 83.7 | $64.8_{\pm0.0}$ |
| GLIPv2-T | 1 | Full | $64.8_{\pm0.6}$ | $18.7_{\pm0.6}$ | $39.5_{\pm1.2}$ | $70.0_{\pm1.5}$ | $70.5_{\pm0.2}$ | $69.8_{\pm18.0}$ | $70.6_{\pm4.0}$ | $68.4_{\pm1.2}$ | $71.0_{\pm1.3}$ | $65.4_{\pm1.1}$ | $68.1_{\pm2.0}$ | $28.9_{\pm2.9}$ | $72.9_{\pm4.7}$ | $52.8_{\pm1.4}$ |
| GLIPv2-T | 3 | Full | $53.9_{\pm0.1}$ | $17.8_{\pm0.7}$ | $42.7_{\pm1.1}$ | $73.1_{\pm1.0}$ | $65.9_{\pm0.2}$ | $84.7_{\pm3.4}$ | $69.7_{\pm0.8}$ | $60.7_{\pm1.3}$ | $28.8_{\pm0.8}$ | $61.7_{\pm1.3}$ | $60.6_{\pm0.2}$ | $35.5_{\pm0.4}$ | $68.3_{\pm1.7}$ | $55.6_{\pm0.7}$ |
| GLIPv2-T | 5 | Full | $58.9_{\pm0.2}$ | $17.4_{\pm1.1}$ | $42.8_{\pm1.3}$ | $72.6_{\pm0.7}$ | $66.1_{\pm0.6}$ | $84.9_{\pm0.9}$ | $69.7_{\pm0.5}$ | $65.5_{\pm1.0}$ | $35.6_{\pm0.9}$ | $62.8_{\pm0.3}$ | $59.8_{\pm0.2}$ | $35.5_{\pm1.2}$ | $74.4_{\pm2.1}$ | $57.4_{\pm0.4}$ |
| GLIPv2-T | 10 | Full | $57.6_{\pm1.0}$ | $27.6_{\pm1.2}$ | $49.1_{\pm1.0}$ | $70.4_{\pm0.5}$ | $69.2_{\pm0.2}$ | $88.1_{\pm0.0}$ | $73.1_{\pm2.3}$ | $58.0_{\pm2.8}$ | $42.9_{\pm1.2}$ | $64.8_{\pm0.2}$ | $62.1_{\pm0.9}$ | $39.9_{\pm0.4}$ | $71.6_{\pm0.8}$ | $59.7_{\pm0.3}$ |
| GLIPv2-T | All | Full | 66.4 | 30.2 | 52.5 | 74.8 | 80.0 | 88.1 | 74.3 | 63.7 | 54.4 | 63.0 | 73.0 | 60.1 | 83.5 | 66.5 |
| GLIPv2-B | 1 | Prompt | $68.7_{\pm0.1}$ | $19.9_{\pm0.3}$ | $38.4_{\pm0.4}$ | $68.5_{\pm1.0}$ | $68.6_{\pm0.8}$ | $87.7_{\pm3.0}$ | $69.3_{\pm1.7}$ | $68.5_{\pm0.4}$ | $55.2_{\pm0.3}$ | $65.7_{\pm0.7}$ | $67.2_{\pm0.1}$ | $34.8_{\pm0.8}$ | $69.6_{\pm0.4}$ | $60.4_{\pm0.3}$ |
| GLIPv2-B | 3 | Prompt | $67.2_{\pm0.6}$ | $22.2_{\pm0.3}$ | $46.5_{\pm0.9}$ | $71.2_{\pm0.8}$ | $70.9_{\pm0.1}$ | $86.9_{\pm0.2}$ | $67.7_{\pm1.8}$ | $63.7_{\pm2.3}$ | $46.9_{\pm0.8}$ | $68.1_{\pm0.4}$ | $67.4_{\pm0.9}$ | $47.9_{\pm1.0}$ | $78.9_{\pm1.7}$ | $62.0_{\pm0.5}$ |
| GLIPv2-B | 5 | Prompt | $68.9_{\pm1.0}$ | $25.7_{\pm0.4}$ | $50.5_{\pm0.9}$ | $73.8_{\pm1.5}$ | $69.7_{\pm0.6}$ | $84.9_{\pm0.3}$ | $69.3_{\pm0.0}$ | $65.8_{\pm1.6}$ | $65.7_{\pm1.0}$ | $69.2_{\pm0.3}$ | $67.5_{\pm0.7}$ | $34.0_{\pm0.2}$ | $73.1_{\pm0.6}$ | $62.9_{\pm0.4}$ |
| GLIPv2-B | 10 | Prompt | $69.4_{\pm0.7}$ | $21.8_{\pm1.3}$ | $48.7_{\pm0.7}$ | $71.3_{\pm0.2}$ | $71.0_{\pm0.7}$ | $88.1_{\pm0.4}$ | $73.5_{\pm0.3}$ | $61.5_{\pm1.9}$ | $69.3_{\pm0.2}$ | $68.6_{\pm0.7}$ | $41.3_{\pm0.7}$ | $75.2_{\pm1.3}$ | $63.8_{\pm0.3}$ | |
| GLIPv2-B | All | Prompt | 71.9 | 26.1 | 50.6 | 74.5 | 73.5 | 86.9 | 74.9 | 71.0 | 71.6 | 71.0 | 72.4 | 50.2 | 80.5 | $67.3_{\pm0.0}$ |
| GLIPv2-B | 1 | Full | $67.8_{\pm0.6}$ | $18.7_{\pm0.3}$ | $44.2_{\pm0.9}$ | $71.4_{\pm0.3}$ | $70.4_{\pm1.2}$ | $87.9_{\pm7.3}$ | $66.1_{\pm2.4}$ | $68.9_{\pm1.1}$ | $60.6_{\pm1.6}$ | $68.1_{\pm0.6}$ | $69.0_{\pm0.7}$ | $35.1_{\pm0.9}$ | $68.9_{\pm2.1}$ | $61.2_{\pm0.6}$ |
| GLIPv2-B | 3 | Full | $68.1_{\pm0.2}$ | $25.7_{\pm0.4}$ | $46.4_{\pm1.6}$ | $69.8_{\pm1.3}$ | $71.3_{\pm1.2}$ | $88.0_{\pm0.3}$ | $68.6_{\pm0.9}$ | $69.8_{\pm1.7}$ | $60.1_{\pm0.3}$ | $68.4_{\pm1.9}$ | $68.5_{\pm0.6}$ | $39.8_{\pm0.6}$ | $71.4_{\pm2.1}$ | $62.8_{\pm0.8}$ |
| GLIPv2-B | 5 | Full | $68.6_{\pm1.0}$ | $21.6_{\pm0.6}$ | $46.7_{\pm0.7}$ | $70.9_{\pm0.9}$ | $71.0_{\pm1.2}$ | $88.1_{\pm3.7}$ | $69.1_{\pm0.2}$ | $71.8_{\pm1.0}$ | $61.5_{\pm0.7}$ | $68.7_{\pm0.2}$ | $69.3_{\pm0.8}$ | $40.2_{\pm1.0}$ | $74.8_{\pm2.8}$ | $63.3_{\pm0.6}$ |
| GLIPv2-B | 10 | Full | $67.4_{\pm1.3}$ | $22.3_{\pm1.1}$ | $46.7_{\pm0.5}$ | $74.3_{\pm0.4}$ | $73.4_{\pm1.1}$ | $85.5_{\pm0.1}$ | $74.7_{\pm0.9}$ | $65.8_{\pm2.4}$ | $53.7_{\pm1.1}$ | $67.4_{\pm0.9}$ | $68.9_{\pm0.7}$ | $52.3_{\pm0.6}$ | $83.7_{\pm2.2}$ | $64.6_{\pm0.3}$ |
| GLIPv2-B | All | Full | 71.1 | 32.6 | 57.5 | 73.6 | 80.0 | 88.1 | 74.9 | 68.2 | 70.6 | 71.2 | 76.5 | 58.7 | 79.6 | 69.4 |
| GLIPv2-H | 1 | Prompt | $68.3_{\pm0.6}$ | $16.4_{\pm0.6}$ | $45.8_{\pm0.3}$ | $72.0_{\pm0.5}$ | $67.9_{\pm0.9}$ | $89.3_{\pm3.2}$ | $69.3_{\pm1.7}$ | $67.9_{\pm0.6}$ | $66.3_{\pm1.9}$ | $68.0_{\pm0.7}$ | $66.8_{\pm0.3}$ | $33.9_{\pm0.4}$ | $70.7_{\pm1.5}$ | $61.4_{\pm0.5}$ |
| GLIPv2-H | 3 | Prompt | $69.5_{\pm0.7}$ | $25.9_{\pm0.2}$ | $50.0_{\pm1.2}$ | $75.4_{\pm1.4}$ | $70.1_{\pm0.9}$ | $85.9_{\pm2.5}$ | $69.3_{\pm0.7}$ | $70.8_{\pm1.2}$ | $66.4_{\pm0.8}$ | $68.0_{\pm1.2}$ | $68.0_{\pm0.7}$ | $34.0_{\pm0.3}$ | $72.7_{\pm1.6}$ | $63.6_{\pm0.6}$ |
| GLIPv2-H | 5 | Prompt | $69.4_{\pm0.7}$ | $22.0_{\pm0.6}$ | $49.1_{\pm0.1}$ | $70.7_{\pm1.0}$ | $73.0_{\pm0.5}$ | $88.1_{\pm0.8}$ | $70.3_{\pm0.4}$ | $71.2_{\pm1.8}$ | $62.9_{\pm1.4}$ | $70.1_{\pm0.3}$ | $68.3_{\pm0.6}$ | $42.7_{\pm0.6}$ | $74.3_{\pm0.5}$ | $63.9_{\pm0.7}$ |
| GLIPv2-H | 10 | Prompt | $66.0_{\pm0.2}$ | $27.5_{\pm1.3}$ | $53.8_{\pm0.2}$ | $74.6_{\pm0.2}$ | $80.1_{\pm0.7}$ | $87.4_{\pm0.4}$ | $69.3_{\pm0.7}$ | $66.0_{\pm0.3}$ | $51.2_{\pm1.9}$ | $67.2_{\pm0.2}$ | $72.8_{\pm0.7}$ | $58.3_{\pm0.2}$ | $76.5_{\pm1.3}$ | $65.5_{\pm0.6}$ |
| GLIPv2-H | All | Prompt | 71.2 | 31.1 | 57.1 | 75.0 | 79.8 | 88.1 | 68.6 | 68.3 | 59.6 | 70.9 | 73.6 | 61.4 | 78.6 | $69.1_{\pm0.0}$ |
| GLIPv2-H | 1 | Full | $67.8_{\pm0.6}$ | $17.3_{\pm0.6}$ | $50.7_{\pm0.4}$ | $63.8_{\pm0.5}$ | $67.3_{\pm0.9}$ | $89.4_{\pm3.2}$ | $69.3_{\pm1.7}$ | $68.2_{\pm0.6}$ | $66.6_{\pm1.9}$ | $66.8_{\pm0.7}$ | $67.0_{\pm0.3}$ | $34.0_{\pm0.4}$ | $75.0_{\pm1.5}$ | $61.7_{\pm0.5}$ |
| GLIPv2-H | 3 | Full | $62.3_{\pm0.2}$ | $29.1_{\pm0.4}$ | $52.8_{\pm1.6}$ | $72.7_{\pm1.3}$ | $78.4_{\pm1.2}$ | $85.8_{\pm3.4}$ | $68.6_{\pm0.9}$ | $69.9_{\pm1.7}$ | $72.2_{\pm0.6}$ | $65.8_{\pm0.8}$ | $68.0_{\pm0.7}$ | $55.9_{\pm0.8}$ | $81.1_{\pm2.1}$ | $64.1_{\pm0.8}$ |
| GLIPv2-H | 5 | Full | $66.4_{\pm1.0}$ | $23.4_{\pm0.6}$ | $50.7_{\pm0.7}$ | $73.9_{\pm0.9}$ | $71.8_{\pm1.2}$ | $84.2_{\pm3.7}$ | $71.2_{\pm0.2}$ | $68.1_{\pm1.0}$ | $67.4_{\pm0.7}$ | $70.8_{\pm0.2}$ | $65.8_{\pm0.8}$ | $54.6_{\pm1.0}$ | $75.6_{\pm2.8}$ | $64.4_{\pm0.6}$ |
| GLIPv2-H | 10 | Full | $67.3_{\pm1.3}$ | $31.6_{\pm1.1}$ | $52.4_{\pm0.7}$ | $71.3_{\pm0.4}$ | $80.0_{\pm1.4}$ | $88.1_{\pm0.1}$ | $72.9_{\pm0.6}$ | $56.9_{\pm2.4}$ | $52.2_{\pm1.1}$ | $65.4_{\pm0.9}$ | $73.9_{\pm0.7}$ | $61.0_{\pm0.6}$ | $84.0_{\pm3.2}$ | $65.9_{\pm0.3}$ |
| GLIPv2-H | All | Full | 74.4 | 36.3 | 58.7 | 77.1 | 79.3 | 88.1 | 74.3 | 73.1 | 70.0 | 72.2 | 72.5 | 58.3 | 81.4 | 70.4 |

Table 6: Per-dataset performance of DyHead, GLIP-T, GLIP-L, and GLIPv2-T, GLIPv2-B and GLIPv2-H. For PascalVOC, we report the mAP (IoU=0.50:0.95) using the COCO evaluation script, to be consistent with other 12 datasets. "Prompt" denotes prompt tuning. "Full" denotes full-model tuning.