# OpenReview forum: "GLIPv2: Unifying Localization and Vision-Language Understanding "
_NeurIPS.cc/2022/Conference — NeurIPS 2022 Accept_

### Official Review · Reviewer_WQgW · 2022-07-11

**Rating:** 6
**Confidence:** 3
**Soundness:** 3 good
**Presentation:** 3 good
**Contribution:** 3 good

**Summary:**

This paper presents a new V&L model, GLIPv2, that builds upon the original GLIP v1 model centered around multimodal pretraining. GLIPv2 aims to unify visual localization tasks (e.g. object detection) and uses V&L understanding tasks like visual question answering. To do so, three distinct pretraining tasks are used 1) phrase grounding, where the model computes the alignment between image regions and tokens 2) and 3) the standard masked language modeling task. Notably, task-specific classification heads are not used for pretraining. In finetuning on downstream tasks, GLIPv2 performs competitively. The model can also be used in zero-shot and prompt-tuned settings.

**Questions:**

*Questions*

- Mentioned in the previous section as well, is the text transformer from a pretrained CLIP model (L239)?
- What are the authors thoughts about requiring another large, multimodal model to generate pretraining data?

*Suggestions*

- About half of the prior work section provides details about GLIPv2; space can be saved by compressing some of the details of this paper. For instance L75-78, which is about half of the localization models subsection, describe the novel contributions of this work instead of prior work and how GLIPv2 differs.
- This sentence (“See GLIP (36) for more 132 details”) is essentially repeated twice in Section 3.1; one can be removed to save space and for readability.

*Typos*

- L103: output -> outputs
- caption Figure 2: compute -> computes
- L118: extract -> extracts
- L128: classier -> classifier
- L139: tyeps -> types
- L206: one set of weight -> one set of weights
- L207: task -> tasks
- L210: keep -> keeps
- L229: broken link to figure


**Limitations:**

- Section 5 is titled “Conclusion and Social Impacts” without a description of social impacts.

- Any biases learned by GLIP (v1) – for instance, having significantly lower detection accuracy for certain demographics of people in images or performing worse for images where people do not fit in the gendered roles of the training data – propagates to the pretraining data for GLIPv2 as well. This is compared to detection datasets that were hand-annotated. It’s a component of using large-scale data, and particularly using large scale data labeled by another ML model, that the authors should address this in their limitations section.

**Strengths And Weaknesses:**

**Strengths**

- Similar to CLIP, GLIPv2 can perform open-vocabulary tasks because of the classification-to-matching trick that computes the dot product between the fused visual and linguistic representations. This means the model is particularly adaptable compared to many other V&L models because it can handle new and out-of-domain visual classes.
- The zero-shot and prompt-tuning experiments have impressive results and are, in and of themselves, exciting to see as tasks for a large V&L model. The ability to evaluate models on downstream tasks with either little or no parameters updates and remain comparable to full finetuning shows the model has learned a lot during pretraining.

**Weaknesses**

- GLIPv2 uses GLIP (v1) to generate bounding boxes for the unlabeled (image, text) pairs in the pretraining data versus just an off-the-shelf object detector that has not had linguistic supervision. This also clouds the data and pretraining approach a bit given how similar GLIPv2 is to GLIPv1.

- The text transformer appears to use the text transformer from CLIP including its pretrained weights (let me know if I'm misunderstanding). If it does use those weights, this skews these results because CLIP's text transformer already had visual supervision from the contrastive pretraining.

---

> ### Author Response · Authors · 2022-08-02
> **Our Response to Reviewer WQgW**
>
> We appreciate the reviewer for the positive and insightful feedback. Our response to the reviewer’s questions is as follows.
> ***
> 1. Mentioned in the previous section as well ("The text transformer appears to use the text transformer from CLIP including its pretrained weights (let me know if I'm misunderstanding). If it does use those weights, this skews these results because CLIP's text transformer already had visual supervision from the contrastive pertaining."), is the text transformer from a pretrained CLIP model (L239)?
>
> **Our Response**: For GLIPv2-T, we use the ImageNet pre-trained Swin-Transformer to initialize the image encoder and BERT-base-uncased to initialize the language encoder. For GLIPv2-B, we use the pre-trained paired image-language encoder from UniCL (CLIP-like pre-training, https://github.com/microsoft/UniCL) for initialization. We did an ablation study on the different language encoders (UniCL vs. BERT) and found that their results are nearly the same. Therefore, UniCL initialization **does not skew** the good localization performance. The main reason for us to keep the UniCL(CLIP-like) language encoder is due to its Pre-LayerNorm (Xiong et al.) operation. We find the UniCL(CLIP-like) language encoder with Pre-LayerNorm is more **stable** during the training compared with BERT, which uses Post-LayerNorm. We will include the ablation study in the revised version.
> - Xiong et al., On Layer Normalization in the Transformer Architecture.
> ***
> 2. GLIPv2 uses GLIP (v1) to generate bounding boxes for the unlabeled (image, text) pairs in the pretraining data versus just an off-the-shelf object detector that has not had linguistic supervision. This also clouds the data and pretraining approach a bit given how similar GLIPv2 is to GLIPv1. "What are the authors thoughts about requiring another large, multimodal model to generate pretraining data?"
>
> **Our Response**: An off-the-shelf object detector cannot generate the bounding boxes and their corresponding phrases for the unlabeled image-text pairs because the traditional object detector **cannot be used as an open-vocabulary grounding model**. GLIP and other grounding models (e.g., MDETR) should be used to generate bounding boxes for the unlabeled image-text pairs. We use the GLIP model to get pseudo labels on Cap and CC+SBU data due to its best grounding performance prior to GLIPv2's work. Furthermore, we can even use GLIPv2 model itself, which is trained on human-annotated OD and GoldG data, to scale up the pre-training data. This method is self-consistent in terms of the self-training approach to utilize large-scale unlabeled image-text pairs data.
> ***
> 3. (a) Any biases learned by GLIP (v1) – for instance, having significantly lower detection accuracy for certain demographics of people in images or performing worse for images where people do not fit in the gendered roles of the training data – propagates to the pretraining data for GLIPv2 as well. This is compared to detection datasets that were hand-annotated. It’s a component of using large-scale data, and particularly using large-scale data labeled by another ML model, that the authors should address this in their limitations section.
> (b) Section 5 is titled “Conclusion and Social Impacts” without a description of social impacts.
>
> **Our Response**: Thank you for all the reviewers' suggestions! While our paper shows promising results on both object detection and VL understanding tasks, additional analysis of the data and the model is necessary before deploying it in practice because large-scale web data may contain unintended private information, unsuitable images/text, or some bias leakage. We will check more carefully about the generated pseudo data and address this in the "Conclusion and Social Impacts" section.

---

> > ### Comment · Reviewer_WQgW · 2022-08-04
> > **Follow up to 1 & 3**
> >
> > 1. That's exciting to hear that performance is maintained even with a text encoder initialization from text-only pretraining! To make this argument slightly more convincing to readers, I would suggest including the localization ablation and ideally other text-only and vision-only metrics as well time-permitting. It still does in fairness feel less convincing to initialize the text stream of the multimodal model with a text encoder from another multimodal model.
> >
> > 3. Awesome -- looking forward to reading more about this in the updated section! I understand it's a known limitation of using any pretrained vision model; this just warrants very clear disclosure and in particular for labels coming from a model like GLIP that has not undergone (to my knowledge) any fairness evaluation.
> >
> > I'll circle back to 2 in a separate comment later on.

---

> > > ### Author Response · Authors · 2022-08-07
> > > **Thanks for the reply**
> > >
> > > Thank Reviewer WQgW for the reply!
> > > We will include the ablation of text encoder initialization, e.g., text-only pretraining model vs clip/unicl-like multimodal pretrained model, in the final version. As we presented in the rebuttal, their performance are nearly the same.
> > >
> > > For the second point, if the reviewer has any questions/concerns, we are happy to answer.

---

### Official Review · Reviewer_iKXy · 2022-07-11

**Rating:** 6
**Confidence:** 4
**Soundness:** 3 good
**Presentation:** 3 good
**Contribution:** 3 good

**Summary:**

This work proposes a new VL grounding framework called GLIPv2, which unifies several localization and VL understanding tasks in a same unified interface. It shows that doing pretraining on localization data + image-text pairs with this setup improves downstream model performance on all taska and achieves SOTA performance on most of them. It inroduces an inter sample contrastive loss which improves performance.

**Questions:**

- Table 4, Row 6 .. is the MLM objective trained alongside other objectives? Or is there another step of pretraining here? Appendix section 4 mentions that Row 6 has additional stage of training without MLM. I think it will be beneficial to add more analysis behind the intuition of a second stage of pretraining.
- How does the training time gets impacted with the newly introduced loss?
- Since the authors discuss how the loss differs from CLIP, did they also compare on vision classification tasks?

**Limitations:**

Yes they have but I would encourage them to discuss in detail any potential negative societal impact of their work.

**Strengths And Weaknesses:**

### Strengths
- Proposed work improves SOTA performance on a variety of localization and vision-language understanding tasks
- The paper is well written and structured
- They present both finetuning with individual task specific heads as well as prompt tuning and using same weights for different tasks for zero shot
- The authors do thorough ablation on all the different combinations of the pretraining objectives and dataset combinations



### Weaknesses
- I am concerned about the novelty as the additional loss term is somewhat similar to well known region-word loss applied over full batch in multiple works and same setup as GLIP but showing performance on new vision+language tasks. Although this work shows better performance compared to previous works, I wonder if this is also due to additional data used for pretraining which is the localized data generated by GLIP.
- It seems there is additional localization data generated using GLIP model that is used for training. If so, are the GLIP and GLIPv2 models trained in a teacher/student setup? In fact, how does a GLIP model perform if retrained on this additional data (kind of like self-training)?  Are the GLIP baselines in Table 1 using this additional localization data? If not, then the comparison is a bit unfair.
- Line 1184-185 is unclear to me what the authors are trying to say. I understand the explanation why Inter-image region-word loss is different from CLIP but the methodology is not very clear.
- Table 2 if there is trian-test overlap, I would suggest the authors to remove those results. Authors should have been careful about this while preparing the training dataset to deduplicate the train set against any downstream test-validation sets.

---

> ### Author Response · Authors · 2022-08-02
> **Our Response to Reviewer iKXy**
>
> We appreciate the reviewer for the positive and insightful feedback. Our response is as follows.
> ***
> 1. I am concerned about the novelty as the additional loss term is somewhat similar to well known region-word loss applied over full batch in multiple works and same setup as GLIP but showing performance on new vision+language tasks.
>
> **Our Response**: As far as we know, up to the deadline (05/19/2022) for NeurIPS submission, there are only three published papers (VILD (ICLR22), RegionCLIP (CVPR22), and X-VLM (ICML22)) that have the flavor of "region-word" loss applied over full batch. We discuss the difference between our work and the three aforementioned works in the following: (1) All these three works use **"region-sentence"** loss, i.e., the similarity between a region feature and the [CLS] token of a sentence, instead of true **"region-word"** loss used in GLIPv2. As a result, none of these three works made use of the phrase grounding data, which may contain multiple entities in one sentence during their training. It is the most important point in GLIPv2 to use phrase grounding data and pseudo grounding data to train a unified grounded VL understanding model. (2) GLIPv2 has carefully designed the **positive label propagation** in our inter-image region-word contrastive loss to mitigate the wrong assumption that "every unpaired region-word pair is negative". We discussed the intuition and necessity of positive label propagation (Please refer to our response to Q2 below). As far as we know, no previous work has mentioned this mechanism of positive label propagation before. (3) There are some other differences. For example, in VILD, its ``region-sentence loss" is actually not a contrastive loss over full-batch but a classification loss over a fixed vocabulary per sample (see the definition of $L_{ViLD-text})$. After all, we believe that our inter-image region-word contrastive loss is novel and has a significant difference from previous works. We will include this discussion in the revision.
> ***
> 2. Line 184-185 is unclear to me what the authors are trying to say. I understand the explanation why Inter-image region-word loss is different from CLIP but the methodology is not very clear.
>
> **Our Response**: We introduce the positive label propagation for the inter-image contrastive loss in L184-185. In our inter-image region-word contrastive loss, we **cannot** simply assign all regions and texts coming from unpaired image-text as negative pair, as done in CLIP. Our datasets contain object detection datasets, such as Objects365. As mentioned in L182-183 and Figure 2, if a region of an image from Objects365 is labeled as "person", this "person" region should be a positive pair with all "person" phrases in language queries from other Objects365 images. "Positive label propagation" is such a mechanism to propagate positive labels based on the detection labels, to avoid false negative supervision in region-word contrastive loss.
>
> However, consider two phrases from phrase grounding datasets (Flickr30k-entities), e.g., "a person with a red hat." and "a person wearing a blue shirt.". Even though they have the same phrase "person", each of them carries semantic context that is unique to that image-sentence pair. Therefore, we do not apply "positive label propagation" to grounding-type data, as mentioned in Line 184-185.
> ***
> 3. Since the authors discuss how the loss differs from CLIP, did they also compare on vision classification tasks?
>
> **Our Response**: The proposed region-word contrastive loss is specific for learning **region-level** representations. It is inspired by CLIP's sentence-image contrastive loss, which focuses on **image-level** representations. Thus, we do not view our loss as an improvement over CLIP but rather a much-needed extension to region-level tasks. Thus, we do not compare with CLIP on classification tasks, which only value image-level representations.
> ***
> (Please see below for more responses.)

---

> > ### Author Response · Authors · 2022-08-02
> > **Our Response to Reviewer iKXy**
> >
> > 4. (a) Although this work shows better performance compared to previous works, I wonder if this is also due to additional data used for pretraining which is the localized data generated by GLIP.
> > (b) It seems there is additional localization data generated using GLIP model that is used for training. If so, are the GLIP and GLIPv2 models trained in a teacher/student setup? In fact, how does a GLIP model perform if retrained on this additional data (kind of like self-training)? Are the GLIP baselines in Table 1 using this additional localization data? If not, then the comparison is a bit unfair.
> >
> > **Our Response**: GLIPv2-T does not use additional pre-training data compared to GLIP-T. The pre-training data of GLIP-T / GLIPv2-T consist of two types of data: 1) gold human-annotated data (gold detection data + gold grounding data); 2) image-text pairs with pseudo boxes (Cap4M). The pseudo boxes are coming from a "teacher GLIP-T" model. Thus, GLIP-T reported in the GLIP[36] paper (see Section 4 in the GLIP paper) is already trained in such a "self-training" manner.
> >
> > To account for any implementation differences, we have provided a rigorous comparison between GLIP and GLIPv2 in Table 4. Row 3 ($L\_{loc}$ + $L\_{intra}$) is our re-implementation of GLIP-T (including using "self-training" data); Row 6 is the GLIPv2-T. Row 3 and Row 6 use the exact same pre-training data (the same image-text pairs and the same pseudo boxes).
> > ***
> > 5. Table 4, Row 6 .. is the MLM objective trained alongside other objectives? Or is there another step of pretraining here? Appendix section 4 mentions that Row 6 has additional stage of training without MLM. I think it will be beneficial to add more analysis behind the intuition of a second stage of pre-training.
> >
> > **Our Response**: An additional stage of pre-training is applied for small models (GLIPv2-T and GLIPv2-B) due to limited model capacity. In order to achieve higher performance on both localization and understanding tasks, we find that including all data (even with some noise) and MLM loss in the first stage of pre-training will benefit the model for learning a better representation of both localization and understanding capability. Since the OD tasks require the model with more accurate localization ability, in our 2nd stage of pre-training, we decide to eliminate the MLM loss. The large model (GLIPv2-H)  does not need this additional stage because it has enough capacity to learn both word-region alignment and MLM together in a single stage. We will include this analysis in the revised version.
> > ***
> > 6. How does the training time gets impacted with the newly introduced loss?
> >
> > **Our Response**:We provide the comparison of the training speed for GLIP and GLIPv2 on V100 below. The GLIP-T achieves 1.62 FPS, and GLIPv2-T achieves 1.46 FPS with both inter-image region-word contrastive loss and MLM loss; GLIP-L achieves 0.88 FPS while GLIP-B achieves 0.83 FPS with comparable performance. Introducing these new losses with nearly negligible computational cost but provides extra gain on the performance on both localization and understanding tasks (Table 1\&2, GLIP vs. GLIPv2).

---

### Official Review · Reviewer_GxcW · 2022-07-12

**Rating:** 6
**Confidence:** 4
**Ethics Flag:** Yes
**Soundness:** 4 excellent
**Presentation:** 3 good
**Contribution:** 3 good

**Summary:**

This paper proposes a GLIPv2 model trained on vision-language grounding, where localization and vision-language understanding tasks are reframed through the lens of grounding to have a unified model. Additionally, a new inter image-text token loss is introduced which provides some performance gains.


**Questions:**

Questions
- On L82-84 authors write “Many VL models (e.g., BUTD) (2; 58) rely on a pre-trained localization model as their visual encoder; the downside is the pro-longed “localization->VLP” pre-training pipeline (41; 48; 13; 47; 34; 32; 60; 37; 35). In contrast, GLIPv2 simplifies the pre-training pipeline and enables grounded VL understanding for better interpretability." Doesn’t GLIPv2 use image tags/bbox labels from a pre-trained detector too? Again, paper states “ triplet format (Img, Text, T)” data inputs where T are (box-label) annotations, so don’t we still start with localization to some extent?
- L183-185 “We do not propagate positives to grounding-type texts (natural sentences) because phrases in sentences carry contexts that
185 are unique to that image-sentence pair.” Out of curiosity did you try to include these as positives? If so, what happened? Seems like a thoughtful design choice.

Style and writing comments
- The writing grammar is not consistently correct throughout the paper.
- L56 “world” → word
- L78 bold section should probably be on a new line. Same with L93?
- L93: arriving *at* a…
- L128: “classier” → “classifier”
- L179: “easy” → “easily”
- L229: Figure reference is missing.
- L206: One set of weight*s*, L207 task*s*

**Limitations:**

The authors did not include limitations or negative societal impact discussion.

**Strengths And Weaknesses:**

Strengths
- The losses and unified framework introduced in the paper are sound and an important direction for vision-language work. It is ideal to have a single model capable of performing both grounding and understanding type tasks, and making the vision-language understanding tasks more grounded for interpretability and improved performance.
- The figures are helpful, well made illustrations and there are extensive experiments for the paper’s method to be validated.
- The method does outperform its predecessor GLIP consistently, albeit small gains. But it does have large performance improvements on VL understanding tasks compared to prior work MDETR and others (e.g., PhraseCut, VQA)

Weaknesses
- While there are some performance improvements over GLIP, I don’t see significant method changes other than the inter loss. This loss does not seem to contribute that much (1-2 point improvement compared to the ablations without the loss), and the performance improvement compared to GLIP is quite small.

---

> ### Author Response · Authors · 2022-08-02
> **Our Response to Reviewer GxcW**
>
> We appreciate the reviewer for the positive and insightful feedback. Our response to the reviewer’s questions is as follows.
> ***
> 1. While there are some performance improvements over GLIP, I don’t see significant method changes other than the inter loss. This loss does not seem to contribute that much (1-2 point improvement compared to the ablations without the loss), and the performance improvement compared to GLIP is quite small.
>
> **Our Response**: First of all, the reviewer also agrees that "The method does outperform its predecessor GLIP consistently, albeit small gains on OD. But it does have large performance improvements on VL understanding tasks compared to prior work MDETR and others". Note that GLIP cannot do VL understanding tasks (e.g., Captioning, VQA). GLIPv2 extends GLIP to a unified localization and VL understanding model, which is a significant methodology advance from GLIP. Second, in our ablation study in Table 4, with the same amount of pseudo data added in pre-training, the inter-image word-region contrastive loss achieves consistent gain across both localization and VL understanding tasks. Notice that 1-2 points improvements on OD tasks are not small. Specifically, 1.0 AP improvement on COCO and 2.0 AP increase on LVIS are considered quite significant for OD tasks, especially compared with a nearly SoTA baseline GLIP.
> ***
> 2. On L82-84 authors write “Many VL models (e.g., BUTD) (2; 58) rely on a pre-trained localization model as their visual encoder; the downside is the pro-longed “localization$\to$VLP” pre-training pipeline (41; 48; 13; 47; 34; 32; 60; 37; 35). In contrast, GLIPv2 simplifies the pre-training pipeline and enables grounded VL understanding for better interpretability." Doesn’t GLIPv2 use image tags/bbox labels from a pre-trained detector too? Again, the paper states “ triplet format (Img, Text, T)” data inputs where T are (box-label) annotations, so don’t we still start with localization to some extent?
>
> **Our Response**: The localization information is necessary for both traditional BUTD[2;58] models and GLIPv2 models. The major difference is that the traditional BUTD VL models may require a two-stage pre-training: first pre-train on detection modules, then pre-train on VL understanding (alignment) modules. However, GLIPv2 unifies the detection and VL understanding to become a single "grounded VL understanding" pre-training task.
>
> The "(Img, Text, Boxes)" data used in GLIPv2 pre-training can be just human-annotated data (see Row1\&2 in Table 5), with which GLIPv2 pre-training does not involve any pseudo data from a pre-trained grounding/localization model. In order to achieve the best performance, GLIPv2 uses image-text pair data with pseudo boxes from a pre-trained GLIP model (see Row3-6 in Table 4), which is trained with the same "grounded VL understanding" task but just with smaller data. Thank you for the suggestion, and we will make this clearer in the revised version!
> ***
> 3. L183-185 “We do not propagate positives to grounding-type texts (natural sentences) because phrases in sentences carry contexts that 185 are unique to that image-sentence pair.” Out of curiosity, did you try to include these as positives? If so, what happened? Seems like a thoughtful design choice.
>
> **Our Response**: We did not try it for two reasons: (1) It is reasonable not to do it. Our datasets mainly include two types of data in our training: (a) object detection datasets, and (b) phrase grounding datasets with natural sentences description. As mentioned in L182 and Figure 2, if a region from (a) with "person" is annotated, it should be a positive pair with all "person" phrase in detection-type (a). However, for natural sentences from the grounding type (b), by looking at the below two phrases, e.g., "a person with a red hat." and "a person wearing a blue shirt.". Even though they have the same phrase "person", we believe each of them carries a unique semantic context. Therefore, we apply label propagation for detection-type (a) but not for grounding type (b). (2) Practically, it is difficult to determine two **free-form** phrases in natural sentences to have the same meaning and also hard to align them in the implementation. It is indeed a thoughtful and careful design choice, and the above are reasons why we apply only label propagation to (b) instead of (a).

---

> > ### Comment · Reviewer_GxcW · 2022-08-07
> > **Follow Up**
> >
> > Hi Authors,
> >
> > Thanks for responding to my comments and questions, I just wanted to let you know I have read over your responses. I do not have further questions.

---

### Review · Ethics_Reviewer_tev8 · 2022-08-03

**Recommendation:**

I think it is possible to address the concerns in the current version of the paper. The authors should just add a brief comment/discussion about potential dual use/negative societal impact

**Ethics Review:**

The authors do not explicitly discuss the negative societal impact of their work, as pointed out by one of the reviewers.

Since the paper proposes a method for localization and vision-language understanding, an applied task with a potentially dual use problem, the paper would benefit from a brief comment/discussion about potential dual use/negative societal impact.

---

### Meta-Review · Area_Chair_Gb4t · 2022-08-24

**Recommendation:** Accept
**Confidence:** Certain

**Metareview:**

All three reviewers provided positive reviews and scores for this paper. They were happy to see the strong empirical evaluations and improvements over GLIP, impressed by the zero shot results, and found the new combination of pre-training objectives interesting. A few questions and concerns were brought up by reviewers that had to do with differentiation to the GLIP paper and model. These concerns including novelty in the loss term, tasks accomplished, need for detection boxes at training time, etc were well addressed by the authors. The reviewers also acknowledged that their questions were answered. Given these positive reviews and discussions, I recommend acceptance.

Note to authors: Please address the comments raised by the ethics reviewer in your final manuscript. Thank you.

**Award:**

No

---

### Decision · Program_Chairs · 2022-09-14

Accept